# Alcoholism gender differences in brain responsivity to emotional stimuli

Kayle S Sawyer[1,2,3,4]*, Nasim Maleki[1,5], Trinity Urban[3], Ksenija Marinkovic[6], Steven Karson[7], Susan M Ruiz[2], Gordon J Harris[3,8], Marlene Oscar-Berman[2,9,10]

[1]Psychology Research Service, VA Healthcare System, Boston, United States; [2]Department of Anatomy and Neurobiology, Boston University School of Medicine, Boston, United States; [3]Department of Radiology, Massachusetts General Hospital, Boston, United States; [4]Sawyer Scientific, LLC, Boston, United States; [5]Department of Psychiatry, Massachusetts General Hospital, Boston, United States; [6]Department of Psychology, San Diego State University, San Diego, United States; [7]Department of Computer Science, Dartmouth College, Hanover, United States; [8]3D Imaging Service, Massachusetts General Hospital, Boston, United States; [9]Department of Psychiatry, Boston University School of Medicine, Boston, United States; [10]Department of Neurology, Boston University School of Medicine, Boston, United States

**Abstract** Men and women may use alcohol to regulate emotions differently, with corresponding differences in neural responses. We explored how the viewing of different types of emotionally salient stimuli impacted brain activity observed through functional magnetic resonance imaging (fMRI) from 42 long-term abstinent alcoholic (25 women) and 46 nonalcoholic (24 women) participants. Analyses revealed blunted brain responsivity in alcoholic compared to nonalcoholic groups, as well as gender differences in those activation patterns. Brain activation in alcoholic men ($ALC_M$) was significantly lower than in nonalcoholic men ($NC_M$) in regions including rostral middle and superior frontal cortex, precentral gyrus, and inferior parietal cortex, whereas activation was higher in alcoholic women ($ALC_W$) than in nonalcoholic women ($NC_W$) in superior frontal and supramarginal cortical regions. The reduced brain reactivity of $ALC_M$, and increases for $ALC_W$, highlighted divergent brain regions and gender effects, suggesting possible differences in the underlying basis for development of alcohol use disorders.
DOI: https://doi.org/10.7554/eLife.41723.001

*For correspondence:
kslays@bu.edu

## Introduction

Impaired affect regulation is a primary motive for the use of drugs, including alcohol (*Prescott et al., 2004*; *Vaughan et al., 2012*). Affective processing deficits have been linked to misinterpretation of environmental cues, irregularity in mood, and increased alcohol consumption and may be an underlying factor leading to the development and maintenance of alcohol use disorders (AUD) (*Gilman and Hommer, 2008*; *Thorberg et al., 2009*). However, problem drinkers are a heterogeneous population. While alcohol and other GABAergic agents such as benzodiazepines typically are considered to be depressants because of their ability to decrease anxiety, tension, and inhibition, they also can function as a stimulant, generating feelings of euphoria and well-being (*Gilman et al., 2008*; *Mukherjee et al., 2008*). These effects can be experienced both by men and by women, but the appeal of alcohol for each gender subgroup of problem drinkers may be driven for contrasting reasons (*Buchmann et al., 2010*). For example, on average, women might drink to decrease negative affect, and men might drink to enhance favorable emotional states

**eLife digest** More than 100 million people worldwide are thought to have alcohol use disorder, also known as AUD, alcohol dependence or alcoholism. People who struggle to regulate their emotions tend to consume more alcohol than others. This suggests that impaired emotion processing may increase the risk of developing the disorder.

Most studies of emotion processing in people with alcohol use disorder do not distinguish between men and women. But evidence suggests that men and women process emotions in different ways. Sawyer et al. set out to explore the possible relationships between emotion processing, gender and alcoholism. Four groups of volunteers took part in the study: abstinent men and women with the disorder, and control groups of men and women without a history of alcoholism. Each group contained between 15 and 21 participants. The two abstinent alcoholic groups had not consumed alcohol for at least 21 days. The average length of abstinence was 7 years.

The volunteers viewed a mixture of emotionally charged and neutral images while lying inside a brain scanner. The emotionally charged images were of happy, erotic, gruesome or aversive scenes. Sawyer et al. measured the difference in brain responses to the emotionally charged images versus the neutral ones, and compared this measure across the four groups of participants. Abstinent alcoholic men showed muted brain responses to the emotionally charged images compared to their female counterparts. This effect was seen in brain regions involved in memory, emotion processing and social processing. The same pattern occurred for all four types of emotionally charged image.

Abstinent alcoholic men also showed smaller brain responses to the emotionally charged images than non-alcoholic control men. By contrast, abstinent alcoholic women showed larger brain responses to the emotionally charged images than non-alcoholic control women. This suggests that abstinent alcoholic men and women differ in the way they process emotions. Future studies should investigate whether these differences emerge over the course of abstinence. They should also examine whether these differences might contribute to, or result from, differences in alcohol use disorder between men and women.

DOI: https://doi.org/10.7554/eLife.41723.002

(*Buchmann et al., 2010*; *Buckner et al., 2006*; *Crutzen et al., 2013*; *Oscar-Berman et al., 2014*; *Ruiz and Oscar-Berman, 2015*).

Research unrelated to AUD has indicated that men and women process emotions differently (*Mareckova et al., 2016*; *Proverbio et al., 2009*), and there are differences between men and women in personality disorders and social impairments (*Becker et al., 2017*; *Nixon et al., 2014*; *Oscar-Berman et al., 2009*; *Oscar-Berman et al., 2014*; *Ruiz and Oscar-Berman, 2013*). Women also have been found to display different psychophysiological reactions to emotional stimuli (*Sawyer et al., 2015*) and to be more emotionally expressive than men (*Kring and Gordon, 1998*). Conversely, men on average have an increased tendency to repress emotional responses (*Birditt and Fingerman, 2003*). Additionally, alcoholic women ($ALC_W$) are two to three times more likely to be diagnosed with anxiety and affective disorders than alcoholic men ($ALC_M$), while $ALC_M$ are twice as likely as $ALC_W$ to have antisocial personality disorders (*Merikangas et al., 1996*; *Oscar-Berman et al., 2009*). The presence of gender-specific deficits in emotional regulation may provide insight into what differentially motivates men and women to abuse alcohol (*Erol and Karpyak, 2015*; *Mosher Ruiz et al., 2017*; *Regier et al., 1990*; *Ruiz and Oscar-Berman, 2015*; *Valmas et al., 2014*).

Emotional processing is associated with activity within well-characterized network-based brain circuitries including prefrontal cortex, insula, cingulate cortex, and medial temporal lobe structures including the amygdala (*Davidson et al., 1999*; *Proverbio et al., 2009*). In functional magnetic resonance imaging (fMRI) studies measuring AUD-related abnormal brain responses during emotional processing (*Beck et al., 2009*; *Chanraud-Guillermo et al., 2009*; *Gilman and Hommer, 2008*; *Heinz et al., 2007*), abstinent ALC individuals showed reduced fMRI activation in the amygdala, hippocampus, anterior cingulate, and medial frontal regions in response to viewing stimuli with a negative affective valence, compared to nonalcoholic control (NC) participants (*Marinkovic et al., 2009*;

*Padula et al., 2015*; *Salloum et al., 2007*); in response to viewing stimuli with a positive affective valence, the ALC individuals showed an increase in activation in the anterior cingulate cortex, prefrontal cortex, ventral striatum, and thalamus (*Heinz et al., 2007*). However, little is known about gender-specific persistent influences of alcoholism-related brain activation in response to affective materials, because little research has compared abstinent alcoholic men (ALC$_M$) and women (ALC$_W$) compared to nonalcoholic men (NC$_M$) and women (NC$_W$) (e.g., *Salloum et al., 2007*). Therefore, using fMRI in conjunction with measures of affective judgments, an important aim of the present exploratory study was to address the need for more research in this domain by examining gender differences in the processing of high-arousal emotional stimuli on brain and behavioral responses in ALC men and women compared to NC men and women. The study was designed within a conceptual model of emotional processing adapted from *Halgren and Marinković (1995)*.

According to this model, when an emotionally salient stimulus is perceived, *Emotional Event Integration and Evaluation* takes place, and a response occurs in widespread and focal dynamic cortico-limbic neural networks (*Figure 1—figure supplement 1*). These circuitries embody different functional systems that amalgamate cognitive with feeling aspects of emotions: (1) Attention and orientation to a salient stimulus occurs in insular, anterior cingulate, prefrontal, and posterior parietal cortices. (2) Emotional event appraisal, integration, and evaluation (as influenced by the ongoing emotional context and the perceiver's personality), takes place in posterior cingulate, orbital and medial prefrontal cortex, and other neocortical sites (e.g., fusiform gyrus and superior temporal sulcus), and limbic structures (e.g., hippocampus and amygdala). (3) Volition and decisions, which determine response choice, are generated in cingulate, precentral, premotor, and supplementary cortices.

Using the above model as a guide, we analyzed brain activation and behavioral responses within a psychological task structure aimed to assess the subjective appraisal of valence of specific emotional categories. We chose to do this in order to disentangle how brain activity during the process of evaluation and interpretation of emotional content distinguished ALC from NC groups. To engage the Emotional Integration and Evaluation System, we asked participants to view complex, emotionally meaningful pictures (*aversive*, *erotic*, *gruesome*, *happy* – and *neutral* for comparison), and to rate how the pictures made them feel (*good*, *bad*, or *neutral*). We chose stimuli representing the contrasting valences, because findings from previous research indicated that one or more of those emotional categories were sensitive to deficits in emotional processing by abstinent ALC groups compared to NC groups (*Heinz et al., 2007*; *Marinkovic et al., 2009*; *Padula et al., 2015*; *Salloum et al., 2007*). It should be noted that the behavioral task is not, explicitly, either an emotion judgment task nor an emotion regulation measure. Instead, we expected the data to reflect neural responsivity as indirect measures of emotion processing and/or emotion regulation to a variety of emotionally salient stimuli.

We were especially interested in responses to happy and aversive stimuli, because (a) they have been shown to be sensitive to gender effects in brain activation levels in abstinent ALC participants who viewed faces with different emotional expressions (*Padula et al., 2015*), and (b) abnormalities in the evaluation of aversive stimuli (which are associated with negative feelings such as fear, pain, and stress), play a crucial role in the transition to AUD or alcohol relapse (*Maleki and Oscar-Berman, 2019*; *Witkiewitz et al., 2015*). Whereas brain activation alterations in emotional processes have been studied in relation to AUD (*Beck et al., 2009*; *Chanraud-Guillermo et al., 2009*; *Gilman and Hommer, 2008*; *Heinz et al., 2007*), gender differences have not been explored in depth, and there is a need for more research in this domain (*Nixon et al., 2014*; *Ruiz and Oscar-Berman, 2013*). Therefore, in accordance with the primary aim of the present exploratory study, we sought to determine how gender differences are manifested in the brain networks outlined by our conceptual model (Event Integration and Evaluation). We hypothesized that AUD-related abnormalities in emotional evaluation (i.e., ratings and reaction times) would differ by gender, and these processes would be reflected by gender differences in brain activity during emotional evaluation. Overall, we expected that the same brain regions as in the well-characterized system involved in emotion processing, as described above, would be involved in emotional processes; however, they would not be impacted in the same way for men and women. We hypothesized that ALC$_M$ would show dampened cortico-limbic activation to stimuli from most of the emotional valence categories, thereby reflecting muted affect. For women, we postulated that the pattern of abnormalities associated with AUD would differ

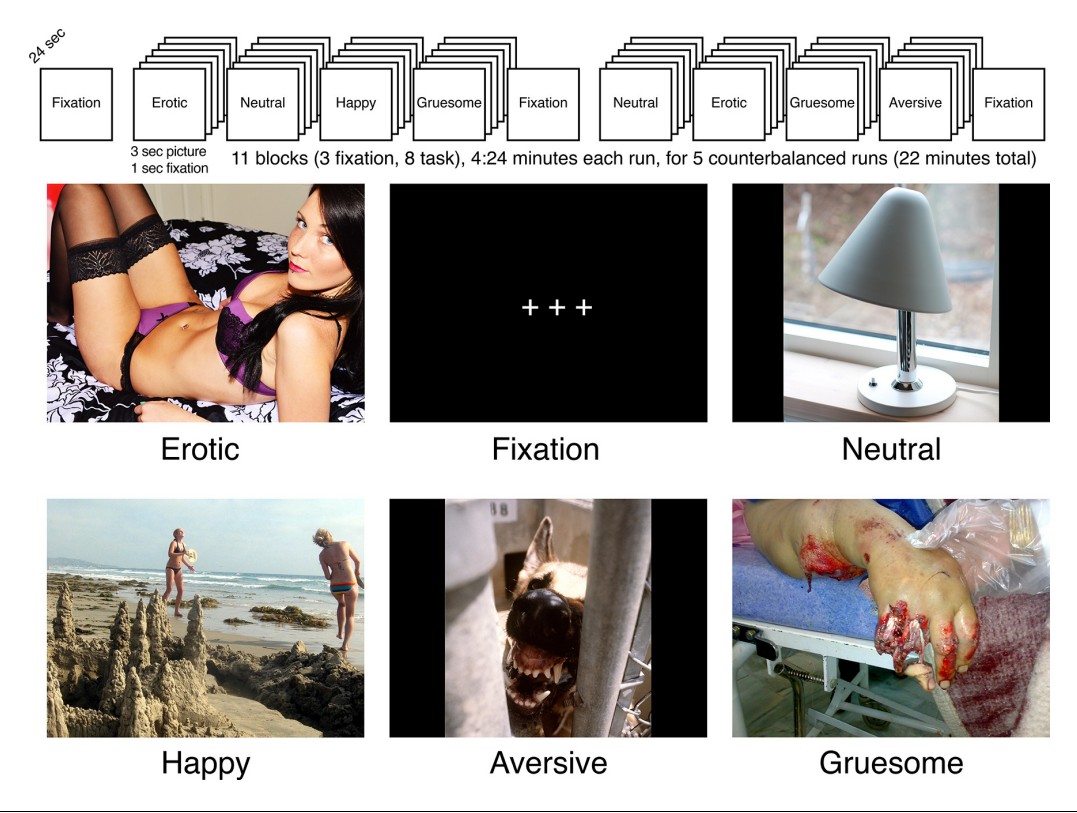

**Figure 1.** Schematic of task presentation, and examples of stimuli. As described in the text, participants were shown pictures from the International Affective Picture System (*Lang et al., 1988*) and asked to report how the pictures made them feel (*good*, *bad*, or *neutral*). Note the pictures in this figure are not the exact pictures shown to participants from the International Affective Picture System as these are not to be made available online (https://csea.phhp.ufl.edu/media/iapsmessage.html). The erotic (https://www.flickr.com/photos/103039225@N05/14964085720) and happy (https://www.flickr.com/photos/moonjazz/2684228420) images are in the public domain and are reproduced here under a Public Domain Mark 1.0 licence (https://creativecommons.org/publicdomain/mark/1.0/). The gruesome (https://commons.wikimedia.org/wiki/File:Amputation_surgery_01.JPG) and neutral (https://commons.wikimedia.org/wiki/File:Herstal_Y1944_med_tiltbar_skjerm-1.JPG) images are in the public domain and are reproduced here under a CC0 1.0 Universal (CC0 1.0) Public Domain Dedication (https://creativecommons.org/publicdomain/zero/1.0/deed.en). The aversive image was taken from the National Archives Catalog (https://catalog.archives.gov/id/6366489) where it was made available with no restrictions on its use.
DOI: https://doi.org/10.7554/eLife.41723.003

The following figure supplement is available for figure 1:

**Figure supplement 1.** Conceptual model of emotional integration and evaluation, adapted from *Halgren and Marinković (1995)*, and informed more recently by results of a meta-analytic analysis by *Riedel et al. (2018)*.
DOI: https://doi.org/10.7554/eLife.41723.004

from that of men, by showing increased activation to emotional stimuli, indicative of hyper-sensitivity to affective input.

## Results

### Participant characteristics

Demographics, alcoholism indices, neuropsychological and clinical assessment scores of the 88 participants are presented in *Figure 2* (and *Appendix 1—tables 1* and *2*). Although the Hamilton Rating Scale for Depression (HRSD; *Hamilton, 1960*) scores for the ALC men and women were higher than for the NC men and women ($p<0.01$), both groups' scores were very low (mean 3.6 vs. 1.1): HRSD scores of 8, 16, and 25 or above indicate mild, moderate, or severe depression, respectively (*Zimmerman et al., 2013*). The average number of daily drinks (DD) was significantly higher in $ALC_M$ compared to $ALC_W$ ($p<0.05$). The alcoholic participants were abstinent for extended lengths, on

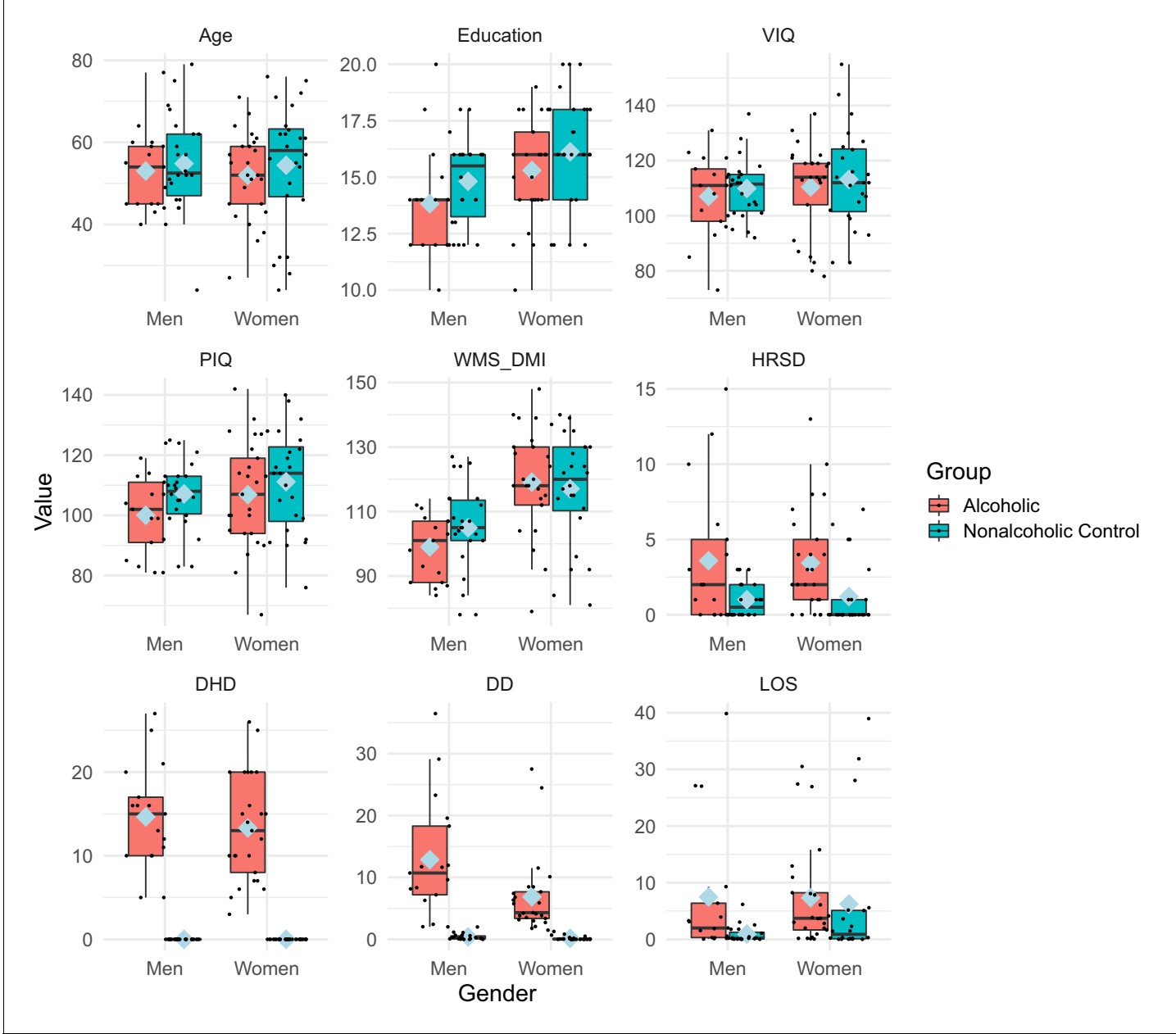

**Figure 2.** Participants' characteristics and drinking measures. The boxplot represents participant characteristics. *Appendix 1—table 1* shows the means, standard deviations, and significant differences. In the boxplot above, blue diamonds indicate mean values. Age, education, DHD, and LOS are expressed in years and DD is in ounces of ethanol per day (approximating daily drinks). LOS values were not applicable for two nonalcoholic control men and four nonalcoholic control women who reported never drinking. Abbreviations: DHD = Duration of Heavy Drinking (>21 drinks per week) in years; DD = Daily drinks; LOS = Length of sobriety in years. HRSD = Hamilton Rating Scale for Depression (*Hamilton, 1960*); VIQ = Wechsler Adult Intelligence Scale, 3rd ed. Verbal Intelligence Quotient; PIQ = Wechsler Adult Intelligence Scale, 3rd ed. Performance Intelligence Quotient; WMS_DMI = Wechsler Memory Scale, 3rd ed. Delayed (General) Memory Index.

DOI: https://doi.org/10.7554/eLife.41723.005

average for seven years. The $ALC_W$ and $NC_W$ had higher delayed memory scores than the $ALC_M$ and $NC_M$ (Wechsler Memory Scale Delayed (General) Memory Index, p<0.01).

## Behavioral ratings

Of the 88 participants included in fMRI analyses, 12 were excluded from the analysis of behavioral ratings because of technical problems or incomplete data, leaving 76 subjects for the final analyses

**Table 1.** Peak voxel or vertex labels of significant clusters for group contrasts of each emotion vs. neutral condition.

Significant clusters (p<0.05 after correction for multiple comparisons) were observed for comparisons between alcoholic and control groups (for the entire sample and for men and women separately), along with group x gender interactions, for each of the four contrasts between each emotion condition compared to the neutral condition. Cortical regions were determined from the peak voxel or vertex. Overall, the table shows that the ALC$_M$ had widespread abnormalities in response to emotional stimuli, and that these effects were significantly different than the effects for the ALC$_W$. Details are described in the text, **Figure 4**, **Figure 5**, and **Figure 6**, and in **Appendix 1—tables 5**, **6** and **7**. Abbreviations: ACC = anterior cingulate cortex; L = left hemisphere; R = right hemisphere; ALC$_W$ = alcoholic women; ALC$_M$ = alcoholic men; NC$_W$ = nonalcoholic control women; NC$_M$ = nonalcoholic control men; ns = not significant; BanksSTS = banks, superior temporal sulcus.

| Lobe | Region at peak | ALC vs. NC | ALC$_W$ vs. NC$_W$ | ALC$_M$ vs. NC$_M$ | Interaction |
|---|---|---|---|---|---|
| Frontal | Caudal Middle Frontal | ns | ns | ns | aversive (L) |
| | Medial Orbitofrontal | ns | ns | ALC$_M$ > NC$_M$: aversive (R) | ns |
| | Rostral ACC | ALC > NC: aversive (L) | ns | ALC$_M$ > NC$_M$: aversive (L) | ns |
| | Rostral Middle Frontal | ns | ns | ALC$_M$ < NC$_M$: happy (R) | happy (L,R), aversive (R) |
| | Precentral | ns | ns | ALC$_M$ < NC$_M$: aversive (L,R), happy (L,R), erotic (R) | aversive (L), happy (L,R), erotic (R) |
| | Superior Frontal | ns | ALC$_W$ > NC$_W$: happy (L) | ALC$_M$ < NC$_M$: aversive (R), erotic (R) | aversive (L), happy (R) |
| | Caudal ACC | ns | ns | ns | happy (L) |
| Parietal | Inferior Parietal | ALC < NC: happy (L) | ns | ALC$_M$ < NC$_M$: aversive (L,R), happy (L) | aversive (L), happy (L) |
| | Postcentral | ALC > NC: erotic (L) | ns | ns | ns |
| | Precuneus | ns | ns | ns | happy (L) |
| | Superior Parietal | ns | ns | ALC$_M$ < NC$_M$: happy (R) | ns |
| | Supramarginal | ns | ALC$_W$ > NC$_W$: aversive (L) | ns | ns |
| Temporal | BanksSTS | ns | ns | ALC$_M$ < NC$_M$: gruesome (L) | gruesome (L) |
| | Parahippocampal | ns | ns | ns | erotic (L) |
| | Cuneus | ns | ns | ns | happy (R) |
| | Pericalcarine | ns | ns | ns | happy (L) |
| Subcortical | Thalamus | ns | ns | ALC$_M$ < NC$_M$: happy (R) | ns |
| Cerebellum | Cerebellum | ns | ns | ns | happy (L), aversive (L) |

DOI: https://doi.org/10.7554/eLife.41723.020

(21 ALC$_W$, 15 ALC$_M$, 21 NC$_W$, 19 NC$_M$). Overall, participants' percentage ratings of *good*, *bad*, and *neutral* were generally consistent among ALC and NC men and women (**Figure 3**) for the various conditions (aversive, erotic, gruesome, happy, neutral). That is, the participants rated erotic pictures as mostly *neutral* and *good*; gruesome pictures as almost entirely *bad*; aversive pictures as *bad*, with a few *neutral*; happy pictures as mostly *good*, with some *neutral*; and neutral pictures as mostly *neutral*, with some *good* (altogether representing a significant condition x rating interaction, **Appendix 1—table 3**). While all groups (ALC and NC men and women) had a similar pattern, a significant group x condition x rating interaction revealed that the ALC group rated erotic pictures as *good* less frequently than the NC group. The gender x condition x rating interaction revealed that more men than women rated erotic pictures as *good*.

As with the percentage ratings, evaluation times also were generally comparable for the four groups (**Figure 3—figure supplement 1** and **Appendix 1—table 4**). There were significant interaction effects of condition x rating, rating x gender, and main effects of condition and rating (p<0.001 for all). The evaluation time for gruesome and aversive stimuli were approximately 0.5 s longer than

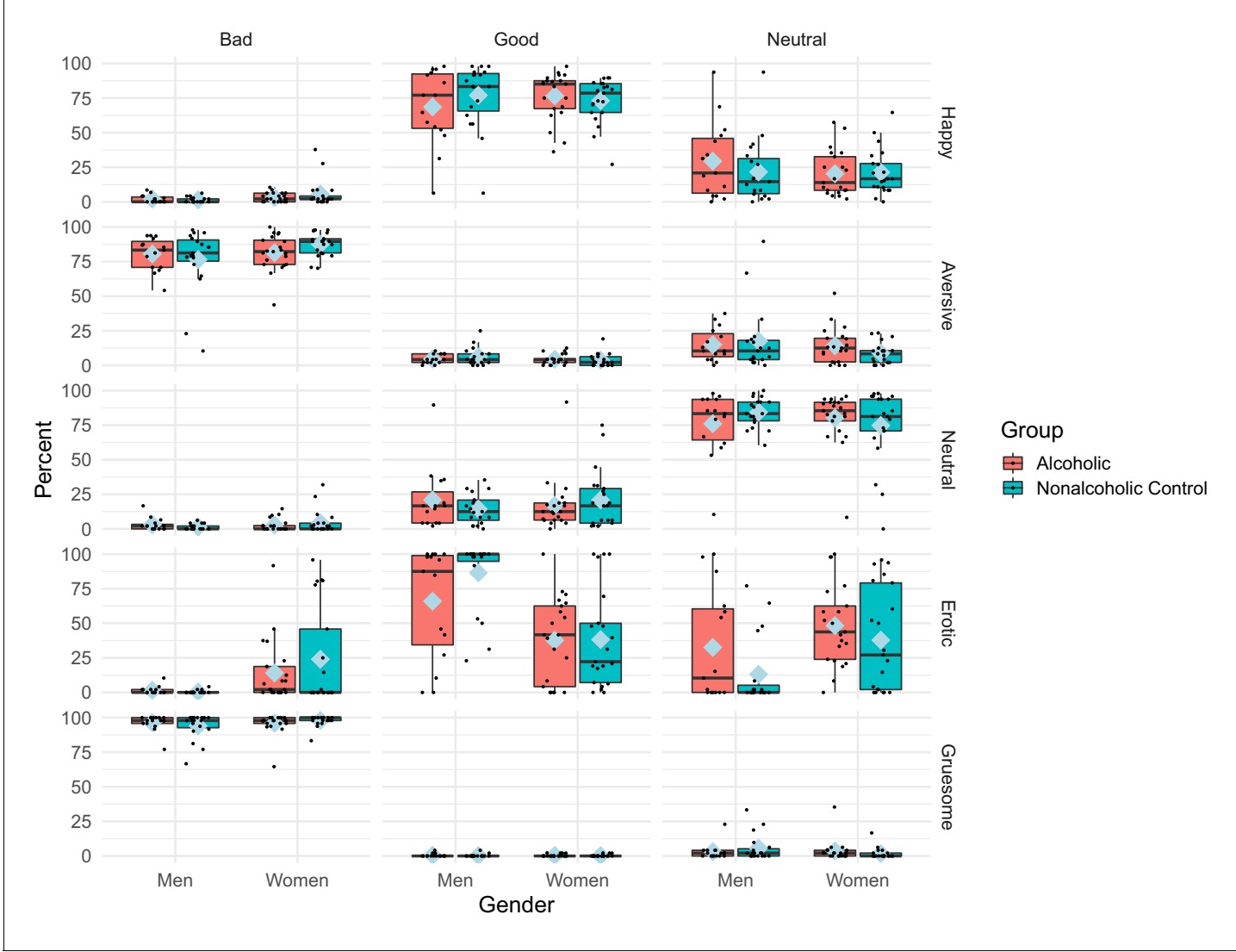

**Figure 3.** Percentage of behavioral ratings by condition, rating, group, and gender. The boxplot represents the significant condition x rating x group interaction, and the significant condition x rating x gender interaction, for percentage rating of the pictures p<0.05 (*Appendix 1—table 3*). The group interaction is most clearly evident for the difference in the *good* and *neutral* ratings of the erotic pictures, with the alcoholic participants rating the pictures *good* less frequently; other picture types were rated more similarly by both the alcoholic and control groups. The gender interaction indicated that men rated erotic pictures as *good* more frequently than women. Blue diamonds indicate mean values. *Figure 3—figure supplement 1* shows the reaction times. Abbreviations: ALC = Alcoholic participants; NC = Nonalcoholic Control participants.

DOI: https://doi.org/10.7554/eLife.41723.006

The following figure supplement is available for figure 3:

**Figure supplement 1.** Reaction times of behavioral ratings by condition, rating, group, and gender.

DOI: https://doi.org/10.7554/eLife.41723.007

other conditions. The evaluation time for *bad* ratings were similarly shorter for gruesome and aversive stimuli. Women took approximately 0.25 s longer (14%) to evaluate the *good* ratings than men, while the evaluation times for *neutral* and *bad* ratings were similar for men and women.

Percentage ratings were significantly predicted by the interaction of Profile of Mood States (POMS) Depression x group x rating, but no post-hoc comparisons were significant after Bonferroni correction. For evaluation times, the following interactions were significant: VIQ x group x gender, and POMS Depression x group x condition x rating, but post-hoc comparisons were not significant for VIQ. The only significant post-hoc group comparison indicated that for the NC group, POMS Depression scores were positively related to evaluation times for *neutral* ratings in the happy

condition (95% confidence interval: [62, 157]), whereas they were not for the ALC group (95% confidence interval: [−19, 40]). In other words, the NC participants with higher Depression scores were slower in rating happy stimuli as being *neutral*.

For the 'caudal middle frontal cluster 1' and 'superior frontal cluster' obtained through analysis of the aversive contrast, percentage ratings were significantly predicted by the interaction of group x gender x rating x contrast effect size. However, post-hoc comparisons of the slopes of contrast effect size for each rating did not identify significant differences between the subgroups. That is, while we identified a different pattern in the relationships of percentage ratings to brain activity among the four subgroups, it was not clear how these relationships differed between the $ALC_W$ vs. $NC_W$, and $ALC_M$ vs. $NC_M$.

## Neuroimaging

The brain activity observed during the neutral condition was subtracted from aversive, erotic, happy, and gruesome conditions, yielding four main comparisons from the study. Overall, the ALC group exhibited lower brain activation values than the NC group for all four contrasts, but significant interactions of group x gender indicated striking differences in these abnormalities. That is, the general observation of lower activation values was evident for $ALC_M$, while $ALC_W$ exhibited a different pattern; the values for each emotion vs. neutral contrast were shifted higher for $ALC_W$. *Table 1* identifies regions with significant group x gender interactions for each of the four contrasts. Because the pattern of these significant group x gender interactions was similar for all contrasts, we have chosen to exemplify the two most salient contrasts: erotic vs. neutral (*Figure 4*) and aversive vs. neutral (*Figure 5*). A summary figure (*Figure 6*) shows the group x gender interactions for all four contrasts.

The contrast of erotic vs. neutral (i.e., erotic minus neutral) is presented in *Figure 4*, which shows that brain activity was greater in most subcortical brain regions for erotic than for neutral images (for $ALC_W$, $ALC_M$, $NC_W$, and $NC_M$). The group x gender interaction revealed a significant cluster that encompassed limbic brain regions including the amygdala, thalamus, hippocampus, and parahippocampal cortex, as well as much of the cerebellum. The erotic and neutral pictures elicited less activation difference for $ALC_M$ than for $NC_M$; this alcoholism-related abnormality was not observed for women.

A complex pattern of gender-related alcoholism abnormalities in brain activity was revealed by the contrast of aversive vs. neutral conditions for several significant clusters (*Figure 5*). For some regions of the brain, activity was higher for aversive than neutral stimuli ('aversive-responding' regions), while for other regions of the brain, activity was higher for neutral than aversive ('neutral-responding' regions). The $ALC_M$ - $NC_M$ comparison resulted in negative values for both aversive-responding and neutral-responding regions, reflecting the following two situations: For aversive-responding regions, the aversive and neutral stimuli had less activation difference for the $ALC_M$ than for the $NC_M$, while for neutral-responding regions, the aversive and neutral stimuli were more similar for $NC_M$ than for the $ALC_M$. In four significant clusters, these negative values obtained from $ALC_M$ were significantly more negative than those obtained from $ALC_W$. As shown in *Figure 5*, three of the clusters were in left prefrontal cortex and one was in the inferior parietal gyrus; similar differences were found for the right hemisphere (*Table 1*). Interestingly, as can be seen in *Table 1*, there also was a significant main effect in two adjoining medial prefrontal regions (medial orbitofrontal and rostral anterior cingulate cortices), wherein alcoholics exhibited higher contrast than controls, and this was more evident in the men than in the women (*Figure 5—figure supplement 1* and *Figure 5—figure supplement 3*). For men, this group difference was in the opposite direction as observed for the regions with significant group x gender interactions.

In summary, we observed a similar pattern of significant group x gender results (*Figure 6*) for each of the four contrasts (aversive, erotic, gruesome, and happy — compared to neutral): $ALC_M$ demonstrated less activation for emotional stimuli compared with neutral images, whereas $ALC_W$ did not show these decreases and in some contrasts, demonstrated activation increases. For comparison with the observations revealed by the erotic contrast shown in *Figure 4* (which highlights the amygdala) and the aversive contrast shown in *Figure 5* (cortical surface), *Figure 6* shows all four of the contrasts, including gruesome and happy.

For $ALC_W$ compared to $NC_W$, significantly more positive brain activation contrasts were seen in superior frontal and supramarginal cortical regions. In $ALC_M$ as compared to $NC_M$, the contrasts revealed more negative values across widespread areas throughout the brain, including the inferior

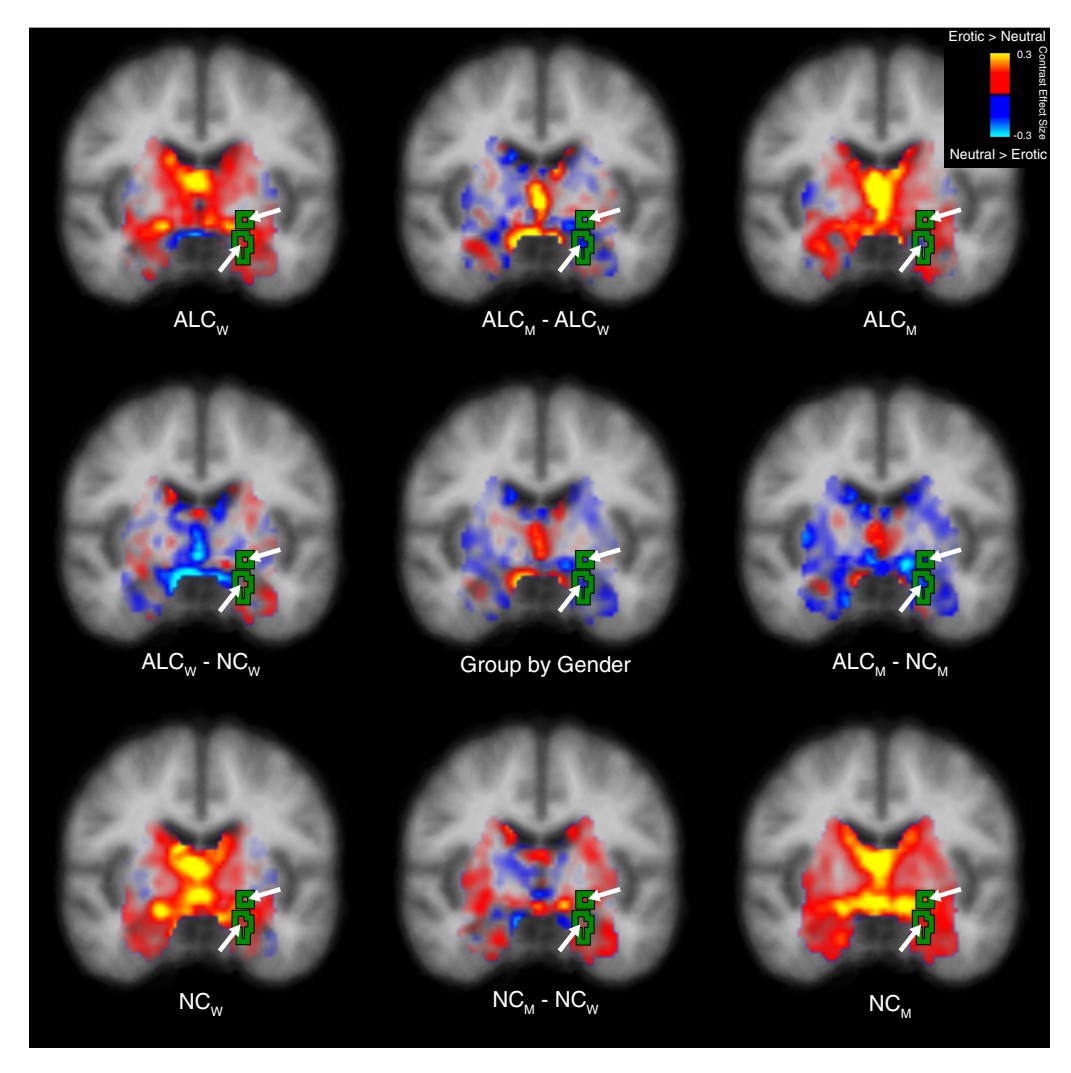

**Figure 4.** Erotic vs. neutral stimuli elicited abnormal activation of the limbic system and cerebellum in alcoholic men. A significant group x gender interaction in response to erotic vs. neutral stimuli was identified and is displayed as a green outline indicated by arrows. All inferior arrows designate the amygdala. Group mean contrast values are displayed in the four brain images located in the corners of the figure, and group comparisons are indicated by minus signs. Contrast values are overlaid on coronal cross sections. Images are displayed in radiological convention with the right hemisphere shown on the left. (Sagittal and axial views are shown in *Figure 4—figure supplement 1* and *Figure 4—figure supplement 2*; *Figure 4—figure supplement 3* shows the magnitude of cluster differences.) Abbreviations: $ALC_M$ = Alcoholic men; $ALC_W$ = Alcoholic women; $NC_M$ = Nonalcoholic men; $NC_W$ = Nonalcoholic women.

DOI: https://doi.org/10.7554/eLife.41723.008

The following figure supplements are available for figure 4:

**Figure supplement 1.** Erotic vs. neutral stimuli elicited abnormal activation of the limbic system and cerebellum in alcoholic men (sagittal view).
DOI: https://doi.org/10.7554/eLife.41723.009

**Figure supplement 2.** Erotic vs. neutral stimuli elicited abnormal activation of the limbic system and cerebellum in alcoholic men (axial view).
DOI: https://doi.org/10.7554/eLife.41723.010

**Figure supplement 3.** Contrast values observed in the cluster for erotic vs. neutral conditions.
DOI: https://doi.org/10.7554/eLife.41723.011

parietal gyrus, anterior cingulate gyrus, and postcentral gyrus (*Table 1* and *Figure 6*). Specifically, significant group x gender interactions were observed in the frontal (superior frontal, rostral and caudal middle frontal), parietal (inferior and superior parietal gyri, and precuneus), and occipital (pericalcarine and cuneus) lobes, as well as the caudal anterior cingulate, parahippocampal gyrus, and cerebellum. Happy and aversive contrasts were especially evident throughout widespread regions;

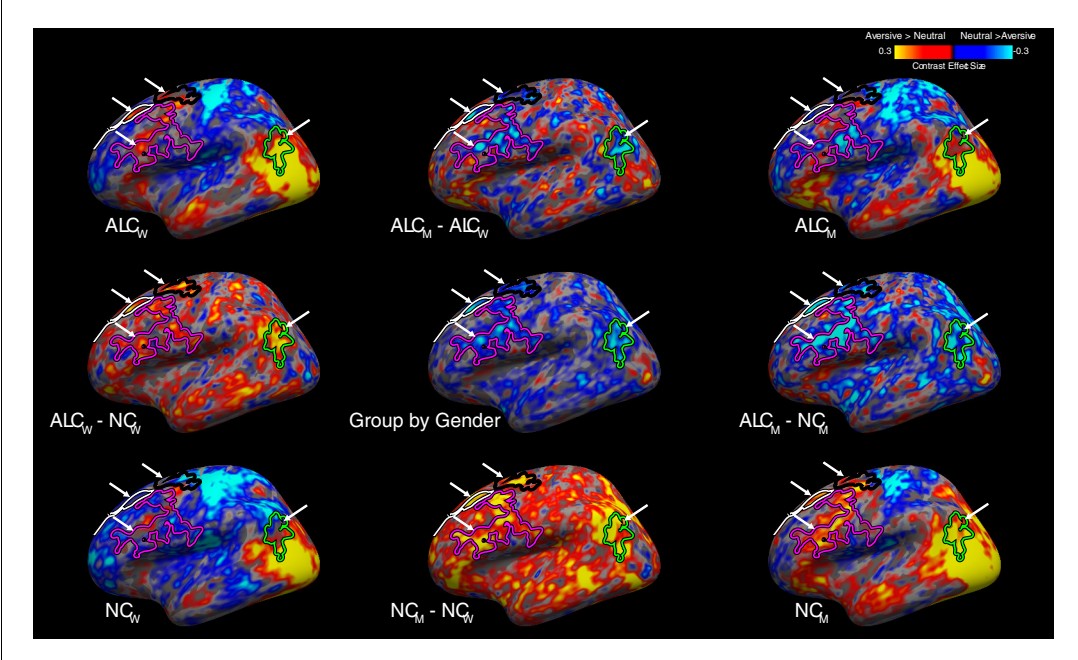

**Figure 5.** Aversive vs. neutral stimuli elicited more abnormally negative responses in alcoholic men. A significant group x gender interaction revealed several clusters (see *Appendix 1—table 6*), which are indicated by arrows on the lateral surface of the left hemisphere, overlaid on contrast values between aversive and neutral stimuli. Group mean contrast values (for aversive vs. neutral) are displayed in the four brain images located in the corners of the figure, and group comparisons are indicated by minus signs. (*Figure 5—figure supplement 1* shows the medial surface, while the right hemisphere is shown in *Figure 5—figure supplement 2* for the lateral and *Figure 5—figure supplement 3* for the medial surface; *Figure 5—figure supplement 4* shows the magnitude of cluster differences.) Although not shown here, the activation patterns across the four subgroups for contrasts of other emotional stimuli (i.e., happy, gruesome, and erotic) with neutral stimuli were similar to those shown above, and likewise, the general locations of the activation regions were similar for the four subgroups. Abbreviations: $ALC_M$ = Alcoholic men; $ALC_W$ = Alcoholic women; $NC_M$ = Nonalcoholic men; $NC_W$ = Nonalcoholic women.

DOI: https://doi.org/10.7554/eLife.41723.012

The following figure supplements are available for figure 5:

**Figure supplement 1.** Aversive vs. neutral stimuli elicited more abnormally negative responses in alcoholic men (left medial surface).
DOI: https://doi.org/10.7554/eLife.41723.013

**Figure supplement 2.** Aversive vs. neutral stimuli elicited more abnormally negative responses in alcoholic men (right lateral surface).
DOI: https://doi.org/10.7554/eLife.41723.014

**Figure supplement 3.** Aversive vs. neutral stimuli elicited more abnormally negative responses in alcoholic men (right medial surface).
DOI: https://doi.org/10.7554/eLife.41723.015

**Figure supplement 4.** Contrast values observed in each cluster for aversive vs. neutral conditions.
DOI: https://doi.org/10.7554/eLife.41723.016

**Figure supplement 5.** Aversive vs.neutral stimuli elicited more abnormally negative responses in alcoholic men (left lateral surface), cluster-forming threshold p<0.001.
DOI: https://doi.org/10.7554/eLife.41723.017

**Figure supplement 6.** Aversive vs.neutral stimuli elicited more abnormally negative responses in alcoholic men (right lateral surface), cluster-forming threshold p<0.001.
DOI: https://doi.org/10.7554/eLife.41723.018

the erotic contrast revealed a significant interaction for limbic structures and cerebellum; and the gruesome contrast revealed an interaction for the superior temporal sulcus.

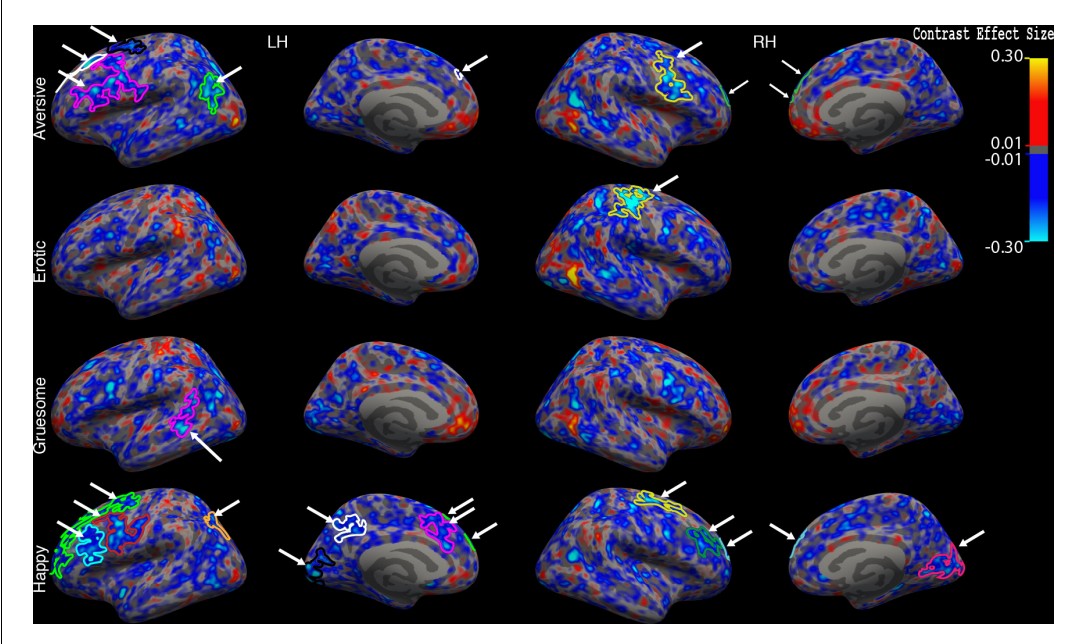

**Figure 6.** Interaction of group x gender for aversive, erotic, gruesome, and happy stimuli vs. neutral stimuli. Significant clusters are indicated by arrows shown on interaction maps of contrast values for each of the four emotions vs. neutral (similar to the center image in *Figure 4* and *Figure 5*). All four brain surfaces are shown (from left: left lateral, left medial, right lateral, and right medial). Blue regions indicate less activation contrast (emotion vs. neutral) for $ALC_M$ relative to $NC_M$ vs. $ALC_W$ relative to $NC_W$. Abbreviations: RH = Right Hemisphere; LH = Left Hemisphere.
DOI: https://doi.org/10.7554/eLife.41723.019

## Discussion

### Alcoholism and emotional processing

Research on the relationship between AUD and emotional dysfunction has shown impairments in self-regulation of emotions, as well as deficits in the perception, identification, evaluation, and understanding of emotions of self and others. However, because little is known about the brain responses to emotional stimuli in $ALC_W$ as compared to $ALC_M$, the present study combined fMRI neuroimaging with a sophisticated experimental design and advanced data analysis methods, to explore the relationship between gender and alcoholism in functional activation of brain regions as participants processed emotional stimuli of varying valences (International Affective Picture System). As indicated in *Table 1*, with the exception of two ventromedial prefrontal regions, our results showed consistently blunted brain activation responses to emotional stimuli vs. neutral stimuli in the ALC group compared to the NC group for men; this general pattern was not observed for women. Further, a significant interaction between gender and alcoholism indicated that the affective pictures elicited lower activation contrasts in $ALC_M$ relative to $NC_M$, abnormalities that were significantly lower and more pervasive than those observed between $ALC_W$ and $NC_W$. That is, by comparison, $ALC_W$ showed more positive activation contrasts than found for $NC_W$, in regions including the superior frontal and supramarginal cortex. In the $ALC_M$, the significant differences appeared in areas throughout the brain, including the inferior parietal gyrus, anterior cingulate gyrus, and postcentral gyrus. *Table 1* (and *Appendix 1—tables 5*, *6* and *7*) and *Figures 4*, *5* and *6* show the extent and spread of the differences among $ALC_M$, $NC_M$, $ALC_W$, and $NC_W$.

### Gender and alcoholism interaction in emotional processing regions in the brain

Emotional processing involves engaging multiple brain regions (*Davidson et al., 1999*). In vivo neuroimaging studies as well as *post-mortem* pathological studies have shown that cortical loss in the frontal lobes is the most common damage observed both in association with AUD (*Oscar-*

*Berman and Marinkovic, 2003*) and in individuals having emotional disorders unrelated to AUD (*Bechara et al., 2000*; *Young et al., 2010*). *Padula et al. (2015)* used fMRI to compare gender effects in affective processing by abstinent alcohol dependent and healthy nonalcoholic individuals. Their stimuli were pictures of individual faces that displayed positive (happy) and negative (sad, fearful) emotional expressions. Similar to our approach, they examined contrasts in activation provoked by the emotion stimuli vs. the neutral stimuli. Of note, our present results are congruent with those reported by *Padula et al. (2015)*, who found significant group x gender interactions in frontal brain activation levels to positive and to negative emotional stimuli. Despite differences in experimental methods, results of both studies are consistent with the notion of gender-specific and alcoholism-related effects in affective processing, with an emphasis on frontal brain involvement.

In our exploratory study, the frontal brain regions showing significant interactions between alcoholism and gender were precentral cortex, rostral and caudal middle frontal cortex, superior frontal cortex, and the caudal anterior cingulate cortex, for both happy and aversive stimuli. Previous fMRI studies have suggested that rostral middle frontal cortex may be involved in the implicit or uninstructed generation and perpetuation of emotional states (*Waugh et al., 2010*; *Waugh et al., 2014*). Moreover, in two studies (*Aldhafeeri et al., 2012*; *Hägele et al., 2016*), the investigators were consistent in their reports of significant increases in prefrontal and amygdala activation levels in response to pleasant and aversive IAPS pictures, respectively (compared to neutral pictures). Given that in our study, $ALC_M$ showed lower activation compared to $NC_M$ in frontal, parietal, and temporal regions in response to most of the categories of emotional stimuli, our findings might reflect deficits in $ALC_M$ in maintaining positive and negative emotions. By comparison, our $ALC_W$ showed higher activation than $NC_W$ in superior frontal cortex in response to happy stimuli, and higher activation in the supramarginal gyrus to aversive stimuli, suggesting possible compensation for deficiency in maintaining positive and negative emotions.

One of the other frontal brain regions that showed a significant gender x alcoholism interaction was the caudal anterior cingulate cortex, a region thought to be involved in appraisal and expression of negative emotion (*Etkin et al., 2011*). However, for the regions anterior to the caudal anterior cingulate, we found a different pattern of group differences. The $ALC_M$ group had greater contrast values than the $NC_M$ group in the subcallosal regions of medial orbitofrontal cortex and rostral anterior cingulate cortex. The difference in the activation of these regions in $ALC_M$ was in the opposite direction to that observed for other regions, where group x gender interactions had been evident. As suggested by our conceptual model of emotional evaluation and integration (*Figure 1—figure supplement 1*), these frontal regions are involved in attending to and integrating cognitive and affective responses to external events (*Bush et al., 2000*; *Margulies et al., 2007*; *Oscar-Berman and Marinković, 2007*; *Riedel et al., 2018*). Therefore, the increased responsivity in the $ALC_M$ group might indicate compensatory involvement in evaluating the emotional pictures (*Oscar-Berman and Marinković, 2007*).

Additionally, significant interactions between gender and alcoholism were observed in cortical regions involved mainly in visual processing, including the cuneus and precalcarine regions, in response to happy stimuli (*Figure 6*). These significant interactions reflect higher contrast values for affective pictures compared to neutral pictures, more so in $NC_M$ than $ALC_M$, whereas the effect was reduced for the two groups of women. In NC participants, we confirmed the greater activation in visual cortex while viewing emotional vs. neutral pictures that has been reported in prior studies, with some suggesting stronger responses by men to pleasant pictures and stronger responses by women to unpleasant pictures (*Sabatinelli et al., 2004*).

Inferior parietal cortex was another region that showed a significant interaction between gender and alcoholism, driven mainly by the blunted activation in the $ALC_M$ compared to the $NC_M$ men. Inferior parietal gyrus is involved in the perception of emotions in facial stimuli (*Sarkheil et al., 2013*). Except for neutral pictures, most of the other pictures had a human face in them, and therefore, the interaction and lower activation in $ALC_M$ may represent an impairment in processing emotional facial expressions. In fact, previous research has shown that long-term abstinent $ALC_M$ showed less activation in temporal limbic areas, when viewing positive or negative emotional faces compared to controls (*Marinkovic et al., 2009*).

There also were significant interactions between gender and alcoholism in limbic and subcortical structures: In $ALC_M$, brain activity for erotic and neutral pictures were relatively similar, leading to decreased differential activation, while $NC_M$ had stronger activity for erotic than neutral pictures, for

parahippocampal cortex, hippocampus, amygdala, other limbic structures, and the cerebellum. This alcoholism-related abnormality was not observed for women: The $ALC_W$ had a slightly larger (although not significant) positive contrast between erotic and neutral pictures compared to $NC_W$.

## Limitations

The results of this exploratory study are to be considered in the context of several limitations. First, our results are based upon cross-sectional data, and as such, it is impossible to determine if chronic alcohol usage caused, or resulted from, the observed dysregulated emotional reactivity, or perhaps a combination of both. Further, these deficits could reflect differences in brain structure that influenced the emotional activity we observed. In that regard, our alcoholic participants were abstinent for extended lengths, on average for seven years, a variable that speaks to the persistent nature of emotion processing deficits in AUD populations. While it remains unclear whether these deficits predate or result from heavy drinking, or whether emotion processing deficits recover over the course of abstinence, a study of accuracy of decoding emotional facial expressions by short- and long-term abstinent alcoholic men and women (*Kornreich et al., 2001*) indicated that deficits in decoding accuracy for anger and disgust, and to a lesser degree sadness, continued with long-term abstinence. Nonetheless, the topic of persistence vs. recovery remains a promising direction for future studies. Second, we had limited information about the potentially confounding variable of smoking status, and therefore, it was not included in the analyses. Smoking abstinence has been associated with increased emotional reactivity in response to unpleasant stimuli (*Versace et al., 2012*) and interactions with alcoholism (*Durazzo et al., 2013*; *Luhar et al., 2013*), and therefore, may have influenced the results of the present study. Third, while there were peak regions of activation differences, these were observed against a background of broad regions identified that were different between each of the emotional conditions and the neutral condition, and the significant group x gender interactions reflected these broad differences in brain activity. We chose not to artificially suppress the display of these widespread effects in our figures by restricting the thresholds. Fourth, the erotic stimuli shown were identical for all participants in order to maintain a consistent experimental paradigm, while at the same time maximizing arousal. To do this, we selected erotic imagery based upon findings from studies measuring arousal levels to erotic stimuli in men and women (*Bradley et al., 2001*; *Israel and Strassberg, 2007*). In those studies, men's behavioral and electrophysiological responses to erotic photographs of women were, on average, much stronger than to erotic photographs of men, whereas responses by women to erotic imagery were similar for photographs of men and women. Therefore, of the 48 erotic pictures presented to the participants in our study, 23 were photographs of women, and 25 were photographs of men and women together. However, participants' sexual orientation was not assessed, and tailoring the photographs to each individual participant might be more effective.

Additionally, previous research (*Glöckner-Rist et al., 2013*) has suggested that direct measures of drinking motives might be helpful in interpreting our findings of gender differences in AUD. In the present study, we did not collect data to assess those variables. However, in a separate sample of abstinent alcoholic men and women with comparable drinking histories and demographic characteristics (*Mosher Ruiz et al., 2017*), we did assess drinking motives, with Cooper's DMQ-R scales (*Cooper, 1994*). Although Cooper's scale is limited in scope, we found that the ALC group scored higher than the NC group on all of the drinking-motives scales, but the interactions between alcoholism, motives for drinking, and gender were not significant.

Finally, as described in the Methods, the *p*-value thresholds used in this study in conjunction with the multiple-comparison cluster correction procedures employed have been shown to have higher false-positive rates than those specified (*Eklund et al., 2016*). This lenient threshold is appropriate in the context of an exploratory study, both because it minimizes the chance of false negatives (Type two error), and also because it allows for the size of the gender effects to be highlighted. However, we additionally conducted analyses with a cluster-forming *p*-value threshold of (p<0.001), which is commonly used for stronger control of false positives (Type one error). The results of those analyses are shown in *Appendix 1—table 8* and *Figure 5—figure supplements 5* and *6*. Two clusters were identified consistent with the group x gender interaction effects highlighted in this exploratory study: left and right lateral frontal clusters for the contrast of aversive vs. neutral.

Despite the above considerations, the findings from the present exploratory study highlight the need for continued research on the overlap between gender differences in processing of emotional stimuli and the development or maintenance of pathological alcohol consumption.

## Conclusions

While blunted emotional reactivity had been observed previously in alcoholics, earlier studies had focused either exclusively on men or had collapsed data across genders (*Gilman et al., 2010*; *Marinkovic et al., 2009*; *Salloum et al., 2007*). Therefore, the present study provides additional insights into emotional processing in alcoholism by examining the influence of gender on brain activation. In our previous studies (*Rivas-Grajales et al., 2018*; *Sawyer et al., 2018*; *Sawyer et al., 2017*; *Sawyer et al., 2016*; *Seitz et al., 2017*), we had reported gender differences in morphometry of cerebral and cerebellar subregions, and white matter integrity, in association with alcoholism history in men and women. In the current study, we reported functional abnormalities in cortical, subcortical, and cerebellar regions involved in emotional processing that were different in alcoholic men and women. Significant interactions between alcoholism and gender in several cortical regions in response to emotional stimuli were observed for the aversive and happy stimuli, as well as large differences between $ALC_M$ and $NC_M$. Areas within the frontal lobes were among the brain regions evidencing the most profound alcoholism-related gender differences.

The brain activity contrasts related to affective vs. neutral stimuli were dampened in $ALC_M$ in the current study, similarly to prior research showing that $ALC_M$ had blunted limbic activation to emotionally expressive faces (*Marinkovic et al., 2009*). Women are traditionally believed to be more emotionally reactive than men (*Merikangas et al., 1996*), and in the current study, whereas $ALC_M$ showed predominately decreased fMRI emotional responsivity, $ALC_W$ had similar or greater brain activity in response to emotional stimuli than $NC_W$, leading to significant group x gender interaction effects. Future prospective research is advised in order to examine gender differences in emotional reactivity and subsequent drinking behavior, to determine the contributions of gender differences that precede AUD, as compared to gender differences that develop as a result of chronic alcoholism.

## Materials and methods

### Participants

Prior to conducting the experiment, we computed estimates of sample size based upon Cohen's *d*, which suggested approximately 20 participants per group were required to detect a medium to large effect size (*Cohen, 1988*), a number confirmed by fMRI-specific research (*Thirion et al., 2007*). A total of 88 participants (25 $ALC_W$, 17 $ALC_M$, 24 $NC_W$, and 22 $NC_M$) were included in the analyses. The characteristics of the participants, including alcoholism indices and neuropsychological test scores are presented in *Figure 2* (and *Appendix 1—tables 1* and *2*) of the Results section; data and code are available from Dryad (https://doi.org/10.5061/dryad.5fn0224) and GitLab (https://gitlab.com/kslays/sawyer-iaps; copy archived at https://github.com/elifesciences-publications/sawyer-iaps). All participants were right-handed English speakers recruited from the Boston, MA (USA) area through flyers placed in facilities and in public places (e.g., churches, stores), and advertisements placed with local newspapers and websites. Selection procedures included an initial structured telephone interview to determine age, level of education, health history, and history of alcohol and drug use.

Specifically, we investigated the stable and persistent sequelae of AUD that are independent of current drinking or withdrawal, by recruiting long-term abstinent participants with a history of heavy drinking. Eligible individuals were invited to the laboratory for further screening and evaluations ranging between five to eight hours over the course of one to three days. Prior to screening, written informed consent was obtained; the protocols and consent forms were approved by the Institutional Review Boards of the participating institutions: Boston University School of Medicine (#H24686), VA Boston Healthcare System (#1017 and #1018), and Massachusetts General Hospital (#2000P001891). Participants were reimbursed $15 per hour for assessments, $25 per hour for scans, and $5 for travel expenses.

Participants underwent medical history interview and vision testing, plus a series of questionnaires (e.g., handedness, alcohol and drug use, HRSD) to ensure they met inclusion criteria. Participants were given the computerized Diagnostic Interview Schedule (*Robins et al., 2000*), which provides lifetime psychiatric diagnoses according to criteria established by the American Psychiatric Association. Participants were excluded from further participation if any source (e.g., hospital records, referrals, or personal interviews) indicated that they had one of the following: Corrected visual acuity worse than 20/50 in both eyes; Korsakoff's syndrome; cirrhosis, major head injury with loss of consciousness greater than 15 min unrelated to AUD; stroke; epilepsy or seizures unrelated to AUD; schizophrenia; HRSD score over 15; electroconvulsive therapy; history of illicit drug use more than once per week within the past five years (except for one $ALC_W$ who had used marijuana more frequently but not during the six months preceding testing, and one $ALC_W$ who had used marijuana once per week for four years, ceasing four years before testing); lifetime history of illicit drug use more than once per week for over 10 years or three times per week for over five years.

Participants received a structured interview regarding their drinking patterns, including length of abstinence and duration of heavy drinking, that is more than 21 drinks per week (one drink: 355 ml beer, 148 ml wine, or 44 ml hard liquor). For each participant, we calculated a Quantity Frequency Index (*Cahalan et al., 1969*), which factors the amount, type, and frequency of alcohol usage (ounces of ethanol per day, roughly corresponding to number of drinks per day) over the last six months (for the NC group), or over the six months preceding cessation of drinking (for the ALC group). The ALC participants met criteria for alcohol abuse or dependence, and had over 21 drinks per week for at least five years in their lifetime; all had abstained from alcohol for at least 21 days. Importantly, to ensure stability in the sequelae of AUD, we investigated long-term abstinent participants with a history of heavy drinking and whose participation was independent of current drinking or withdrawal. None of the NC participants drank heavily (21 or more per week), except for one man who drank while serving in the army decades before the scan, but did not meet the criteria for alcohol dependence; social drinking patterns of the NC participants are reported in *Figure 2* and *Appendix 1— table 1*. We examined the group x gender interaction within a regression model for the demographics, alcoholism indices, neuropsychological and clinical assessment scores. We also conducted Welch's t-tests to examine gender differences for each measure for the ALC and NC groups separately, and group differences for the men and women separately.

## MRI acquisition

Imaging data were acquired using a 3T Siemens (Erlangen, Germany) Trio Tim magnetic resonance scanner. Following automated shimming and scout image acquisition, two eight-minute 3D T1-weighted MP-RAGE sequences were obtained: TR = 2530 msec, TE = 3.45 msec, flip angle = $7^0$, FOV = 256 mm, 128 sagittal slices with in-plane resolution $1 \times 1$ mm, slice thickness = 1.33 mm. These two structural volumes were used for functional slice prescription, spatial normalization, and cortical surface reconstruction. Due to time constraints, only one MP-RAGE sequence was obtained for 23 subjects (11 $NC_M$, 8 $ALC_M$, 2 $NC_W$, 2 $ALC_W$). Functional whole-brain blood oxygen level-dependent (BOLD) images were obtained with a gradient echo T2*-weighted sequence: TR = 2 s, TE = 30 msec, flip angle = $90^0$, FOV = 200 mm, slice thickness = 3.0 mm, spacing = 1.0 mm, 32 interleaved axial-oblique slices aligned to the anterior-commissure/posterior-commissure line (voxel size: $3.1 \times 3.1 \times 4.0$ mm). The scans covered the entire cerebrum and the superior portion of the cerebellum.

## Behavioral task

Participants were presented with blocks of pictures chosen to evoke emotional responses (*Figure 1*). The picture stimuli were from the International Affective Picture System (*Lang et al., 1988*). Participants completed five runs (except one $NC_W$ who completed only four runs), each including five conditions: aversive, erotic, gruesome, happy, and neutral pictures. As depicted in *Figure 1*, each run contained three 24 s blocks of fixation plus eight 24 s blocks that each consisted of six pictures of one of the emotional conditions (e.g., happy pictures), for a total of 11 blocks per run. The five runs included a total of 40 blocks of emotional pictures with eight blocks for each of the five emotional picture conditions. Stimuli were presented only once, totaling 48 pictures per 264 s run (240 pictures in 22 min in total across the five runs).

Within stimulus blocks, the six pictures were each serially presented against a black background for 3 s, followed by 1 s of fixation (+++). Participants were instructed to answer the question: 'How does the picture make you feel?' Following each image within a block, participants indicated feeling *good, bad, or neutral*, by using their index fingers to press buttons on a box. The left index finger indicated *good*, the right index finger indicated *bad*, and both center buttons indicated *neutral*; the left and right were counterbalanced across participants. Block order was counterbalanced across runs, and run order was counterbalanced across participants. The task was presented with the Presentation software package (Neurobehavioral Systems, Albany, CA, USA).

Behavioral response data were analyzed using R software mixed models (*Bates et al., 2015*; *R Development Core Team, 2017*), with one model specified for reaction times, and one model specified for the percentage of pictures endorsed for each rating (*good, bad, neutral*). For both reaction times and percentage models, independent intercepts were modeled for each participant, and full-factorial ANOVAs were calculated for the four factors of rating (*good, bad, neutral*), condition (aversive, erotic, gruesome, happy, neutral), group (ALC, NC), and gender (men, women).

Full-factorial mixed models were employed to examine the relationships of percentage ratings and evaluation times to selected neuropsychological measures (Wechsler Verbal and Performance IQ scores, and the Delayed Memory Index), affective measures (the POMS Depression scale, and the Multiple Affect Adjective Check List [MAACL] Anxiety and Sensation Seeking scales), and brain activity (i.e., contrast effect size) within the clusters identified to have significant group x gender interactions for aversive vs. neutral and erotic vs. neutral contrasts (the two most salient contrasts). Separate mixed models were used for each measure (three neuropsychological measures, three affect measures, and five clusters, for percentage rating and evaluation times, resulting in a total of 22 models). Outliers (outside three standard deviations from the mean) were removed prior to analyses; this resulted in the exclusion of 1 $ALC_W$ and 1 $ALC_M$ for POMS Depression, and 2 $ALC_W$ and 1 $NC_W$ for MAACL Anxiety. Models were examined for significant ($p<0.05$) interactions of the measures with group or gender, and followed by planned comparisons: ALC vs. NC for group interactions, and subgroup differences ($ALC_W$ vs. $NC_W$, $ALC_M$ vs. $NC_M$) for group x gender interactions. Post-hoc comparisons examined the slope of each measure with percentage ratings or evaluation times, and Bonferroni correction was applied for the number of contrasts examined within the model.

## MRI analyses

The imaging data were analyzed using FreeSurfer and FS-FAST v6.0 (http://surfer.nmr.mgh.harvard.edu) analysis packages (*Dale et al., 1999*; *Fischl et al., 1999a*). Individual cortical surfaces were reconstructed using automatic gray and white matter segmentation, tessellation, and inflation. Images were registered with a canonical brain surface (fsaverage) based on sulcal and gyral patterns (*Fischl et al., 1999b*), and registered with a canonical brain volume (MNI305) using a 12 degrees of freedom nonlinear transform. Gray and white matter surface accuracy was individually examined using automatically-generated quality control figures (https://github.com/poldracklab/niworkflows), and no errors were detected for any of the subjects included in the analyses that would be likely to influence the outcomes of this project (*Waters et al., 2018*).

The fMRI data were corrected for motion and slice-time acquisition using FS-FAST preprocessing. Normalized motion and signal intensity spikes were obtained from the nipype rapidart algorithm (https://www.nitrc.org/projects/rapidart/, https://doi.org/10.5281/zenodo.596855), and blocks with motion over 1.5 mm, or signal intensity shifts over 3.0 standard deviations, were removed via a paradigm file covariate for each run. Subjects were removed from the study if this process excluded all but two or fewer blocks of any condition, a requirement that resulted in the exclusion of two additional $NC_W$. Next, the FS-FAST process split the analysis into three spaces (left and right surfaces, and subcortical volume), then data from each subject was spatially normalized (co-registered with) the fsaverage and MNI305 spaces, respectively; all subsequent analyses were performed in these three group spaces. Spatial smoothing was performed with a 5 mm full width at half maximum Gaussian kernel in 3D for the volume and in 2D for the surfaces. Condition-specific effects were estimated by fitting the amplitudes of boxcar functions convolved with the FSL canonical hemodynamic response function to the BOLD signal across all runs.

Statistical maps were constructed from each contrast of stimulus conditions for each subject (first level analyses). Four contrasts were examined: aversive vs. neutral, happy vs. neutral, erotic vs.

neutral, and gruesome vs. neutral. These first-level analyses were concatenated, and second-level (group level or between-subjects) analyses were performed using random-effects models to account for inter-subject variance (*Friston et al., 1999*), with weighted least squares effects incorporated from the variability measures from the first-level contrasts. We examined the overall main effect of group (ALC vs. NC), the interaction of group x gender, and the effects of group for men and women separately, for each of the four contrasts (each emotion condition vs. neutral condition). Cluster-level corrections for multiple comparisons were applied to cortical surface statistical contrast maps (*Hagler et al., 2006*) using 10,000 precomputed Z Monte Carlo simulations and applied to subcortical volumetric statistical contrast maps using gaussian random fields with a cluster forming threshold of $p<0.05$ and a cluster-wise threshold of $p<0.05$ (further corrected to $p<0.017$ for the analysis of three spaces: left cortex, right cortex, and subcortical). While these procedures have been shown to have a false positive (Type one error) level higher than the one specified (*Eklund et al., 2016*), the present exploratory study was designed to reveal the sizes of the effects, and balance minimizing the chance of a false negative (Type two error) with the goal of highlighting the broad regions where further investigation of gender differences may be warranted. Therefore, the *p*-value threshold was set to a value sufficiently liberal to achieve this goal. For comparisons with research using stricter *p*-values, we additionally conducted the same analyses using a cluster-forming threshold of $p<0.001$, the results of which are discussed in the Limitations. Cortical surface cluster regions were identified by the location of each cluster's peak vertex on the cortical surface (*Desikan et al., 2006*), and subcortical cluster regions were identified by the MNI coordinates of each cluster's peak voxel (*Fischl et al., 2002*).

## Acknowledgements

This research was supported by US Department of Veterans Affairs Clinical Science Research and Development (I01CX000326); National Institute on Alcohol Abuse and Alcoholism (NIAAA) of the National Institutes of Health, US Department of Health and Human Services (R01AA07112, R01AA016624, K05AA00219, and K01AA13402); Athinoula A Martinos Center for Biomedical Imaging Shared Instrumentation Grants (1S10RR023401, 1S10RR019307, and 1S10RR023043); Alcoholic Beverage Medical Research Foundation; Mental Illness and Neuroscience Discovery (MIND) Institute; NIH National Center for Research (P41RR14075); and Boston University Clinical and Translational Sciences Institute (BU CTSI; 1UL1TR001430). We gratefully thank Elinor Artsy, Sheeva Azma, Howard Cabral, Doug Greve, Julie Howard, Sharon Jaffin, Yohan John, Chris Markiewicz, Diane Merritt, EmilyKate McDonough, Alan Poey, Daniel Salz, Yulia Spantchak, Maria Valmas, and Robert Zondervan for help with recruitment assistance, materials, data collection, consultation, and analysis. We further appreciate the suggestions provided by the reviewers and editors at eLife. The content is solely the responsibility of the authors and does not necessarily represent the official views of the National Institutes of Health, the U.S. Department of Veterans Affairs, or the United States Government.

## Additional information

### Competing interests

Kayle S Sawyer: Is an employee of Sawyer Scientific, LLC. There are no other competing interests to declare. The other authors declare that no competing interests exist.

### Funding

| Funder | Grant reference number | Author |
| --- | --- | --- |
| U.S. Department of Veterans Affairs | I01CX000326 | Marlene Oscar-Berman |
| National Institute on Alcohol Abuse and Alcoholism | R01AA07112 | Marlene Oscar-Berman |
| ABMRF/The Foundation for Alcohol Research | | Ksenija Marinkovic |

| National Institute on Alcohol Abuse and Alcoholism | R01AA016624 | Ksenija Marinkovic |
|---|---|---|
| National Institute on Alcohol Abuse and Alcoholism | K05AA00219 | Marlene Oscar-Berman |
| National Institute on Alcohol Abuse and Alcoholism | K01AA13402 | Ksenija Marinkovic |

The funders had no role in study design, data collection and interpretation, or the decision to submit the work for publication.

### Author contributions

Kayle S Sawyer, Data curation, Software, Formal analysis, Supervision, Investigation, Visualization, Writing—original draft, Project administration, Writing—review and editing; Nasim Maleki, Formal analysis, Supervision, Writing—original draft, Writing—review and editing; Trinity Urban, Resources, Supervision, Investigation, Writing—original draft, Project administration; Ksenija Marinkovic, Conceptualization, Resources, Supervision, Funding acquisition, Methodology, Project administration, Writing—review and editing; Steven Karson, Data curation, Software, Formal analysis; Susan M Ruiz, Data curation, Project administration; Gordon J Harris, Marlene Oscar-Berman, Conceptualization, Supervision, Funding acquisition, Project administration, Writing—review and editing

### Author ORCIDs

Kayle S Sawyer (iD) https://orcid.org/0000-0001-7767-5688
Ksenija Marinkovic (iD) https://orcid.org/0000-0003-1658-4496
Marlene Oscar-Berman (iD) http://orcid.org/0000-0003-0100-8883

### Ethics

Human subjects: Prior to screening, written informed consent was obtained; the protocols and consent forms were approved by the Institutional Review Boards of the participating institutions: Boston University School of Medicine (#H24686), VA Boston Healthcare System (#1017 and #1018), and Massachusetts General Hospital (#2000P001891). Participants were reimbursed $15 per hour for assessments, $25 per hour for scans, and $5 for travel expenses.

### Decision letter and Author response

Decision letter https://doi.org/10.7554/eLife.41723.034
Author response https://doi.org/10.7554/eLife.41723.035

## Additional files

### Supplementary files
• Transparent reporting form
DOI: https://doi.org/10.7554/eLife.41723.021

### Data availability

The characteristics of the participants, including alcoholism indices and neuropsychological test scores are presented in Figure 2 (and supplements) of the Results section; data and code are available from Dryad (https://doi.org/10.5061/dryad.5fn0224) and GitLab (https://gitlab.com/kslays/sawyer-iaps; copy archived at https://github.com/elifesciences-publications/sawyer-iaps).

The following dataset was generated:

| Author(s) | Year | Dataset title | Dataset URL | Database and Identifier |
|---|---|---|---|---|
| Sawyer KS, Maleki N, Urban T, Ksenija M | 2018 | Data from: Alcoholism Gender Differences in Brain Responsivity to Emotional Stimuli | https://dx.doi.org/10.5061/dryad.5fn0224 | Dryad, 10.5061/dryad.5fn0224 |

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

# Appendix 1

DOI: https://doi.org/10.7554/eLife.41723.022

**Appendix 1—table 1. Participants' characteristics and drinking measures.** Values presented as mean ± standard deviation. Abbreviations: $ALC_W$ = Alcoholic women; $ALC_M$ = Alcoholic men; $NC_W$ = Nonalcoholic control women; $NC_M$ = Nonalcoholic control men; DHD = Duration of Heavy Drinking (>21 drinks per week) in years; DD = Daily drinks; LOS = Length of sobriety in years. HRSD = Hamilton Rating Scale for Depression (**Hamilton, 1960**); VIQ = Wechsler Adult Intelligence Scale, 3rd ed. Verbal Intelligence Quotient; PIQ = Wechsler Adult Intelligence Scale, 3rd ed. Performance Intelligence Quotient; WMS DMI = Wechsler Memory Scale, 3rd ed. Delayed (General) Memory Index. Significant differences: [a]($ALC_M > NC_M$, $p<0.05$); [b]($ALC_M > NC_M$, $p<0.001$); [c]($ALC_W > NC_W$, $p<0.01$); [d]($ALC_W > NC_W$, $p<0.001$); [e]($ALC_W > ALC_M$, $p<0.001$); [f]($ALC_M > ALC_W$, $p<0.05$); [g]($NC_W > NC_M$, $p<0.05$); [h]($NC_W > NC_M$, $p<0.01$); [i](group x gender interaction, $p<0.05$). [j]LOS values were not applicable for two $NC_M$ and four $NC_W$ who reported never drinking.

| Measure | $ALC_W$ N = 25 | | | $ALC_M$ N = 17 | | | $NC_W$ N = 24 | | | $NC_M$ N = 22 | | |
|---|---|---|---|---|---|---|---|---|---|---|---|---|
| Age | 52.0 | ± | 10.6 | 53.2 | ± | 9.7 | 54.4 | ± | 15.4 | 55.0 | ± | 12.4 |
| Education[g] | 15.3 | ± | 2.3 | 13.8 | ± | 2.5 | 16.1 | ± | 2.6 | 14.8 | ± | 1.9 |
| VIQ | 110.4 | ± | 16.6 | 107.0 | ± | 15.0 | 113.2 | ± | 17.8 | 109.9 | ± | 11.1 |
| PIQ | 106.9 | ± | 17.7 | 100.1 | ± | 12.5 | 111.2 | ± | 16.9 | 107.1 | ± | 11.8 |
| WMS DMI[e,h] | 119.1 | ± | 15.9 | 99.0 | ± | 10.3 | 117 | ± | 17.3 | 105.0 | ± | 14.0 |
| HRSD[a,c] | 3.4 | ± | 3.5 | 3.6 | ± | 4.7 | 1.2 | ± | 2.2 | 1.0 | ± | 1.2 |
| DHD[b,d] | 13.3 | ± | 6.4 | 14.6 | ± | 6.2 | 0.0 | ± | 0.0 | 0.0 | ± | 0.0 |
| DD[b,d,f,i] | 6.9 | ± | 6.3 | 12.9 | ± | 9.6 | 0.2 | ± | 0.3 | 0.4 | ± | 0.5 |
| LOS[j] | 7.3 | ± | 8.9 | 7.5 | ± | 11.9 | 6.2 | ± | 11.8 | 0.9 | ± | 1.5 |

DOI: https://doi.org/10.7554/eLife.41723.023

**Appendix 1—table 2. Neuropsychological and affect scores for alcoholic men and women.** Values presented as mean ± standard deviation. Abbreviations: $ALC_W$ = alcoholic women; $ALC_M$ = alcoholic men; $NC_W$ = nonalcoholic control women; $NC_M$ = nonalcoholic control men; Abbreviations: FSIQ = Wechsler Adult Intelligence Scale, 3rd ed. Full Scale Intelligence Quotient; WMS IMI = Wechsler Memory Scale, 3rd ed. Immediate Memory Index; WMS WMI = Wechsler Memory Scale, 3rd ed. Working Memory Index; POMS = Profile of Mood States (**McNair, 1971**); MAACL = Multiple Affective Adjective Checklist (**Zuckerman and Lubin, 1985**). Significant differences: [a]($ALC_M > NC_M$, $p<0.05$); [b]($ALC_W > NC_W$, $p<0.05$); [c]($ALC_W > ALC_M$, $p<0.05$); [d]($ALC_W < ALC_M$, $p<0.05$); [e]($ALC_M < ALC_W$, $p<0.001$); [f]($NC_M > NC_W$, $p<0.05$); [g]($NC_M < NC_W$, $p<0.05$); [h]($NC_M < NC_W$, $p<0.01$).

| Measure | $ALC_W$ N = 25 | | | $ALC_M$ N = 17 | | | $NC_W$ N = 24 | | | $NC_M$ N = 22 | | |
|---|---|---|---|---|---|---|---|---|---|---|---|---|
| FSIQ | 109.7 | ± | 17.2 | 104.3 | ± | 13.5 | 113.5 | ± | 17.6 | 109.4 | ± | 10.3 |
| WMS IMI[e,h] | 117.4 | ± | 17.4 | 94.8 | ± | 11.0 | 116.2 | ± | 17.5 | 102.9 | ± | 14.2 |
| WMS WMI | 103.9 | ± | 15.9 | 104.6 | ± | 11.3 | 110.2 | ± | 15.0 | 102.9 | ± | 10.3 |
| POMS Tension[a,b] | 38.7 | ± | 9.3 | 39.1 | ± | 6.4 | 33.6 | ± | 6.3 | 34.9 | ± | 5.9 |
| POMS Depression[a,b,f] | 38.9 | ± | 8.2 | 42.4 | ± | 7.0 | 34.5 | ± | 4.2 | 37.5 | ± | 4.1 |
| POMS Anger[a,b] | 43.1 | ± | 6.8 | 44.5 | ± | 7.3 | 39.8 | ± | 3.5 | 39.9 | ± | 4.0 |
| POMS Vigor[b] | 59.4 | ± | 11.3 | 60.8 | ± | 7.7 | 66.3 | ± | 10.2 | 61.9 | ± | 7.7 |

*Appendix 1—table 2 continued*

| Measure | ALC$_W$ N = 25 | | | ALC$_M$ N = 17 | | | NC$_W$ N = 24 | | | NC$_M$ N = 22 | | |
|---|---|---|---|---|---|---|---|---|---|---|---|---|
| POMS Fatigue | 44.1 | ± | 9.2 | 46.4 | ± | 8.5 | 41.0 | ± | 6.6 | 42.7 | ± | 5.6 |
| POMS Confusion[b] | 41.3 | ± | 8.2 | 42.3 | ± | 7.6 | 36.7 | ± | 5.2 | 38.6 | ± | 6.6 |
| MAACL Anxiety | 51.6 | ± | 17.0 | 47.1 | ± | 11.8 | 44.3 | ± | 13.0 | 44.1 | ± | 6.8 |
| MAACL Depression | 56.9 | ± | 26.6 | 57.4 | ± | 32.2 | 47.5 | ± | 12.9 | 46.8 | ± | 7.7 |
| MAACL Hostility | 49.4 | ± | 12.3 | 45.3 | ± | 6.2 | 46.9 | ± | 13.0 | 43.5 | ± | 3.2 |
| MAACL Positive Affect[c,g] | 62.2 | ± | 8.4 | 57.6 | ± | 4.6 | 64.9 | ± | 7.0 | 60.6 | ± | 6.9 |
| MAACL Sensation Seeking[d] | 51.4 | ± | 6.8 | 49.1 | ± | 8.1 | 55.5 | ± | 6.8 | 50.3 | ± | 6.8 |
| MAACL Dysphoria | 53.8 | ± | 22.8 | 47.6 | ± | 17.4 | 44.7 | ± | 16.3 | 42.7 | ± | 6.5 |
| MAACL Positive Affect Sensation Seeking[c,f] | 59.7 | ± | 8.0 | 55.5 | ± | 4.7 | 63.3 | ± | 6.6 | 58.5 | ± | 6.3 |

DOI: https://doi.org/10.7554/eLife.41723.024

**Appendix 1—table 3. Analysis of variance for percentage of pictures rated.** Abbreviations: DF = degrees of freedom. Significance codes: ***p<0.001; *p<0.05

| | Sum of squares | Mean square | Numerator DF | Denominator DF | F | p-value | |
|---|---|---|---|---|---|---|---|
| Condition | 0 | 0 | 4 | 1080 | 0 | 1.00 | |
| Rating | 16152 | 8076 | 2 | 1080 | 24.47 | 4.04E-11 | *** |
| Group | 0 | 0 | 1 | 1080 | 0 | 1.00 | |
| Gender | 0 | 0 | 1 | 1080 | 0 | 1.00 | |
| Condition x Rating | 1208715 | 151089 | 8 | 1080 | 457.84 | 2.20E-16 | *** |
| Condition x Group | 0 | 0 | 4 | 1080 | 0 | 1.00 | |
| Rating x Group | 1880 | 940 | 2 | 1080 | 2.85 | 0.06 | |
| Condition x Gender | 0 | 0 | 4 | 1080 | 0 | 1.00 | |
| Rating x Gender | 8326 | 4163 | 2 | 1080 | 12.62 | 3.84E-06 | *** |
| Group x Gender | 0 | 0 | 1 | 1080 | 0 | 1.00 | |
| Condition x Rating x Group | 5200 | 650 | 8 | 1080 | 1.97 | 0.02 | * |
| Condition x Rating x Gender | 34694 | 4337 | 8 | 1080 | 13.14 | 2.20E-16 | *** |
| Condition x Group x Gender | 0 | 0 | 4 | 1080 | 0 | 1.00 | |
| Rating x Group x Gender | 1525 | 762 | 2 | 1080 | 2.31 | 0.10 | |
| Condition x Rating x Group x Gender | 4762 | 595 | 8 | 1080 | 1.8 | 0.07 | |

DOI: https://doi.org/10.7554/eLife.41723.025

**Appendix 1—table 4. Analysis of variance for reaction times of pictures rated.** Abbreviations: DF = degrees of freedom. Significance codes: ***p<0.001

| | Sum of squares | Mean square | Numerator DF | Denominator DF | F | p-value |
|---|---|---|---|---|---|---|

*Appendix 1—table 4 continued*

|  | Sum of squares | Mean square | Numerator DF | Denominator DF | F | p-value |  |
|---|---|---|---|---|---|---|---|
| Condition | 2614420 | 653605 | 4 | 746.22 | 4.843 | 7.34E-04 | *** |
| Rating | 7536931 | 3768465 | 2 | 750.69 | 27.921 | 2.01E-12 | *** |
| Group | 46807 | 46807 | 1 | 80.48 | 0.347 | 0.56 |  |
| Gender | 119106 | 119106 | 1 | 80.48 | 0.882 | 0.35 |  |
| Condition x Rating | 49431086 | 6178886 | 8 | 744.53 | 45.779 | 2.20E-16 | *** |
| Condition x Group | 540297 | 135074 | 4 | 746.22 | 1.001 | 0.41 |  |
| Rating x Group | 428401 | 214201 | 2 | 750.69 | 1.587 | 0.20 |  |
| Condition x Gender | 900660 | 225165 | 4 | 746.22 | 1.668 | 0.15 |  |
| Rating x Gender | 2152290 | 1076145 | 2 | 750.69 | 7.973 | 3.75E-04 | *** |
| Group x Gender | 3721 | 3721 | 1 | 80.48 | 0.028 | 0.87 |  |
| Condition x Rating x Group | 1230435 | 153804 | 8 | 744.53 | 1.14 | 0.33 |  |
| Condition x Rating x Gender | 1950493 | 243812 | 8 | 744.53 | 1.806 | 0.07 |  |
| Condition x Group x Gender | 286187 | 71547 | 4 | 746.22 | 0.53 | 0.71 |  |
| Rating x Group x Gender | 220441 | 110221 | 2 | 750.69 | 0.817 | 0.44 |  |
| Condition x Rating x Group x Gender | 1248592 | 156074 | 8 | 744.53 | 1.156 | 0.32 |  |

DOI: https://doi.org/10.7554/eLife.41723.026

**Appendix 1—table 5. Cortical brain activation differences between alcoholic and nonalcoholic control participants.** MNI305 coordinates for peak voxel within significant clusters of activation showing difference between alcoholic and nonalcoholic control participants determined by surface-based whole brain analyses in (a) all subjects, (b) women only, and (c) men only. Abbreviations: LH = left hemisphere; RH = right hemisphere; Max = maximum −log10(p-value) in the cluster; VtxMax = vertex number at the maximum; size = surface area of cluster; XYZ = the MNI coordinates of the maximum; CWP = clusterwise p-value further corrected for the three spaces of left cortex, right cortex, and volume; CWPLow and CWPHi = 90% confidence interval for CWP; NVtxs = number of vertices in the cluster; ALC = alcoholic participants; NC = nonalcoholic Control participants.

| Structure | Max | VtxMax | Size (mm$^2$) | X | Y | Z | CWP | CWPLow | CWPHi | NVtxs | Contrast | Comparison |
|---|---|---|---|---|---|---|---|---|---|---|---|---|
| A. All Participants |  |  |  |  |  |  |  |  |  |  |  |  |
| Inferior Parietal Gyrus (LH) | −3.307 | 104494 | 876.17 | −40.6 | −76.1 | 21.6 | 0.00180 | 0.00090 | 0.00270 | 1560 | happy | ALC < NC |
| Rostral Anterior Cingulate (LH) | 3.497 | 37787 | 784.55 | −6.6 | 24.1 | −9.7 | 0.01106 | 0.00867 | 0.01344 | 1397 | aversive | ALC > NC |
| Postcentral Gyrus (LH) | 4.460 | 29054 | 798.38 | −48.8 | −25.1 | 47.1 | 0.01046 | 0.00838 | 0.01284 | 1852 | erotic | ALC > NC |
| B. Women |  |  |  |  |  |  |  |  |  |  |  |  |
| Superior Frontal Gyrus (LH) | 4.392 | 73009 | 838.05 | −7.4 | 39.1 | 30.0 | 0.00659 | 0.00479 | 0.00838 | 2333.2 | happy | ALC$_W$ > NC$_W$ |

*Appendix 1—table 5 continued on next page*

*Appendix 1—table 5 continued*

| Structure | Max | VtxMax | Size (mm²) | X | Y | Z | CWP | CWPLow | CWPHi | NVtxs | Contrast | Comparison |
|---|---|---|---|---|---|---|---|---|---|---|---|---|
| Supramarginal-Gyrus (LH) | 3.399 | 6263 | 662.10 | −51.5 | −52.8 | 25.6 | 0.03469 | 0.03058 | 0.03879 | 1457 | aversive | $ALC_W > NC_W$ |
| **C. Men** | | | | | | | | | | | | |
| Inferior Parietal (LH) | −4.829 | 117735 | 640.53 | −26.7 | −63.2 | 34.5 | 0.04287 | 0.0385 | 0.04724 | 1366 | aversive | $ALC_M < NC_M$ |
| Inferior Parietal (LH) | −4.124 | 47811 | 1371.62 | −40.1 | −75 | 22.2 | 0.0003 | 0 | 0.0006 | 2633 | happy | $ALC_M < NC_M$ |
| Inferior Parietal Gyrus (RH) | −3.35 | 157480 | 916.21 | 30.7 | −63 | 39.6 | 0.00389 | 0.0027 | 0.00539 | 1717 | aversive | $ALC_M < NC_M$ |
| Inferior Parietal (RH) | −4.212 | 68984 | 714.12 | 44.1 | −57 | 14.7 | 0.02322 | 0.01997 | 0.02646 | 1489 | aversive | $ALC_M < NC_M$ |
| Medial Orbito-frontal (RH) | 3.771 | 125131 | 874.65 | 11.8 | 45.8 | −4.1 | 0.00509 | 0.0036 | 0.00659 | 1476 | aversive | $ALC_M > NC_M$ |
| Precentral (LH) | −4.165 | 30289 | 1617.73 | −40.8 | 0.9 | 27.5 | 0.0003 | 0 | 0.0006 | 3248 | aversive | $ALC_M < NC_M$ |
| Precentral (LH) | −4.782 | 66552 | 1845.52 | −39.5 | 1.4 | 26.3 | 0.0003 | 0 | 0.0006 | 3529 | happy | $ALC_M < NC_M$ |
| Precentral (RH) | −4.265 | 60264 | 808 | 51.2 | 3.9 | 30.9 | 0.01046 | 0.00838 | 0.01284 | 1668 | aversive | $ALC_M < NC_M$ |
| Precentral (RH) | −2.771 | 118687 | 831.46 | 23.6 | −6.5 | 46.4 | 0.00927 | 0.00718 | 0.01136 | 1810 | happy | $ALC_M < NC_M$ |
| Precentral (RH) | −3.442 | 92562 | 1459.97 | 40.3 | −9.3 | 60 | 0.0003 | 0 | 0.0006 | 3278 | erotic | $ALC_M < NC_M$ |
| Rostral Anterior Cingulate (LH) | 3.937 | 117327 | 739.44 | −6.3 | 33.3 | −7.8 | 0.01789 | 0.01493 | 0.02085 | 1373 | aversive | $ALC_M > NC_M$ |
| Rostral Middle Frontal (RH) | −3.615 | 116765 | 1340.96 | 33.6 | 30.1 | 32.8 | 0.0003 | 0 | 0.0006 | 2268 | happy | $ALC_M < NC_M$ |
| Rostral Middle Frontal (RH) | −4.17 | 103943 | 775.78 | 22.4 | 62.2 | 2 | 0.01284 | 0.01046 | 0.01522 | 996 | happy | $ALC_M < NC_M$ |
| Superior Frontal (RH) | −5.827 | 93897 | 1018.4 | 25.8 | 24.1 | 38.9 | 0.0009 | 0.0003 | 0.0015 | 2033 | aversive | $ALC_M < NC_M$ |
| Superior Frontal (RH) | −3.971 | 35035 | 639.33 | 17.7 | 56.2 | 17.3 | 0.04636 | 0.04171 | 0.05101 | 977 | aversive | $ALC_M < NC_M$ |
| Superior Frontal (RH) | −3.718 | 35035 | 774.08 | 17.7 | 56.2 | 17.3 | 0.01374 | 0.01106 | 0.01641 | 1212 | erotic | $ALC_M < NC_M$ |
| Superior Parietal (RH) | −4.14 | 74265 | 791.49 | 24.9 | −77.7 | 34.2 | 0.01136 | 0.00897 | 0.01374 | 1313 | happy | $ALC_M < NC_M$ |
| Banks, Superior Temporal Sulcus (LH) | −3.174 | 27674 | 637.52 | −55.7 | −46 | −1.4 | 0.04375 | 0.03937 | 0.0484 | 1387 | gruesome | $ALC_M < NC_M$ |

DOI: https://doi.org/10.7554/eLife.41723.027

**Appendix 1—table 6. Cortical brain activation regions corresponding to the interactions between gender and alcoholism.** MNI305 coordinates for peak voxel within significant clusters of activation showing group x gender interaction for emotion (happy, aversive, gruesome, and erotic vs. neutral) from surface-based, and volumetric whole brain analyses. Abbreviations: LH = left hemisphere; RH = right hemisphere; Max = maximum −log10(*p*-value) in the cluster; VtxMax = vertex number at the maximum; Size = surface area of cluster; XYZ = Montreal Neurological Institute (MNI) coordinates of the maximum; CWP = clusterwise *p*-value further corrected for the three spaces of left cortex, right cortex, and volume; CWPLow and CWPHi = 90% confidence interval for CWP; NVtxs = number of vertices in the cluster.

| Structure | Max | VtxMax | Size (mm²) | X | Y | Z | CWP | CWPLow | CWPHi | NVtxs | Contrast |
|---|---|---|---|---|---|---|---|---|---|---|---|

*Appendix 1—table 6 continued on next page*

*Appendix 1—table 6 continued*

| Structure | Max | VtxMax | Size (mm²) | X | Y | Z | CWP | CWPLow | CWPHi | NVtxs | Contrast |
|---|---|---|---|---|---|---|---|---|---|---|---|
| Superior Frontal (LH) | −2.349 | 152510 | 675.48 | −19.7 | 6.8 | 57.1 | 0.03293 | 0.02911 | 0.03703 | 1451 | aversive |
| Superior Frontal (RH) | −3.900 | 67995 | 825.33 | 16.4 | 58.5 | 14.1 | 0.00957 | 0.00748 | 0.01165 | 1204 | happy |
| Rostral Middle Frontal (LH) | −4.069 | 3407 | 3086.86 | −25.8 | 47.0 | 15.3 | 0.00030 | 0.00000 | 0.00060 | 5250 | happy |
| Rostral Middle Frontal (LH) | −3.467 | 4907 | 1022.51 | −40.3 | 28.4 | 20.7 | 0.00090 | 0.00030 | 0.00150 | 1907 | happy |
| Rostral Middle Frontal (RH) | −3.254 | 116765 | 984.31 | 33.6 | 30.1 | 32.8 | 0.00210 | 0.00120 | 0.00300 | 1677 | happy |
| Rostral Middle Frontal (RH) | −4.624 | 42522 | 753.58 | 22.9 | 54.4 | 17.0 | 0.01522 | 0.01255 | 0.01789 | 1119 | aversive |
| Caudal Middle Frontal (LH) | −4.250 | 76029 | 2047.63 | −43.1 | 2.8 | 47.2 | 0.00030 | 0.00000 | 0.00060 | 4069 | aversive |
| Caudal Middle Frontal (LH) | −6.084 | 47079 | 957.26 | −27.3 | 21.3 | 36.0 | 0.00180 | 0.00090 | 0.00270 | 1728 | aversive |
| Inferior Parietal (LH) | −3.003 | 68612 | 660.83 | −38.8 | −55.1 | 21.4 | 0.03733 | 0.03323 | 0.04141 | 1423 | aversive |
| Inferior Parietal (LH) | −2.822 | 12076 | 689.53 | −29.4 | −65.2 | 40.4 | 0.02794 | 0.02440 | 0.03146 | 1405 | happy |
| Precentral (LH) | −4.363 | 80254 | 1602.74 | −46.4 | −2.0 | 38.6 | 0.00030 | 0.00000 | 0.00060 | 3292 | happy |
| Precentral (RH) | −4.268 | 26942 | 1245.03 | 50.9 | 3.4 | 31.0 | 0.00030 | 0.00000 | 0.00060 | 2436 | aversive |
| Precentral (RH) | −3.352 | 1446 | 923.39 | 23.5 | −5.5 | 46.5 | 0.00389 | 0.00270 | 0.00539 | 2049 | happy |
| Precentral (RH) | −3.211 | 145233 | 1749.60 | 36.1 | −20.1 | 52.8 | 0.00030 | 0.00000 | 0.00060 | 3917 | erotic |
| Pericalcarine (LH) | −5.885 | 12910 | 1431.93 | −5.0 | −69.9 | 11.4 | 0.00030 | 0.00000 | 0.00060 | 1866 | happy |
| Precuneus (LH) | −2.594 | 69252 | 653.39 | −16.0 | −47.8 | 34.8 | 0.03937 | 0.03498 | 0.04375 | 1285 | happy |
| Cuneus (RH) | −3.366 | 86177 | 1658.66 | 5.4 | −84.3 | 19.6 | 0.00030 | 0.00000 | 0.00060 | 2305 | happy |
| Caudal Anterior Cingulate (LH) | −3.722 | 37463 | 670.38 | −6.7 | 29.1 | 22.1 | 0.03352 | 0.02970 | 0.03762 | 1297 | happy |
| Banks, Superior Temporal Sulcus (LH) | −4.428 | 86543 | 625.34 | −56 | −44.7 | −2.6 | 0.04782 | 0.04316 | 0.05246 | 1361 | gruesome |

DOI: https://doi.org/10.7554/eLife.41723.028

**Appendix 1—table 7. Significant brain activation differences determined through volumetric based comparisons.** MNI305 coordinates for peak voxel within significant clusters of activation determined through volumetric whole brain analyses. Abbreviations: LH = left hemisphere; RH = right hemisphere; Max = maximum −log10(*p*-value) in the cluster; XYZ = Montreal Neurological Institute (MNI) coordinates of the maximum; CWP = clusterwise *p*-value further corrected for the three spaces of left cortex, right cortex, and volume.

| Structure | Size (mm³) | X | Y | Z | CWP | Max | Comparison | Contrast |
|---|---|---|---|---|---|---|---|---|
| Parahippocampal Cortex (LH) | 15920 | −34 | −23 | −27 | 0.0007912 | −5.62445 | Group x Gender | erotic |
| Accumbens (LH) | 50016 | −10 | 5 | -9 | 0.000000 | 5.49738 | Control: male vs. female | erotic |
| Cerebellum Cortex (LH) | 15960 | −14 | −39 | −23 | 0.00020 | −3.790 | Group x Gender | happy |
| Cerebellum Cortex (LH) | 7224 | -8 | −45 | −13 | 0.0467673 | 3.25617 | Control: male vs. female | happy |

*Appendix 1—table 7 continued*

| Structure | Size (mm³) | X | Y | Z | CWP | Max | Comparison | Contrast |
|---|---|---|---|---|---|---|---|---|
| Thalamus (RH) | 12384 | 6 | −19 | 11 | 0.0016394 | −3.93561 | Male: alc vs. control | happy |
| Cerebellum Cortex (LH) | 18648 | −22 | −79 | −29 | 0.0000831 | 4.05785 | Control: male vs. female | aversive |
| Cerebellum Cortex (LH) | 15240 | -6 | −41 | −19 | 0.00049 | −3.409 | Group x Gender | aversive |

DOI: https://doi.org/10.7554/eLife.41723.029

**Appendix 1—table 8. Brain activation clusters identified using a cluster forming threshold of p<0.001.** MNI305 coordinates for peak voxel within significant clusters of activation for emotion (happy, aversive, gruesome, and erotic vs. neutral) from surface-based and volumetric whole brain analyses. Abbreviations: LH = left hemisphere; RH = right hemisphere; Max = maximum −log10(*p*-value) in the cluster; VtxMax = vertex number at the maximum; Size = surface area of cluster; XYZ = Montreal Neurological Institute (MNI) coordinates of the maximum; CWP = clusterwise *p*-value further corrected for the three spaces of left cortex, right cortex, and volume; CWPLow and CWPHi = 90% confidence interval for CWP; NVtxs = number of vertices in the cluster; NC = nonalcoholic control group; ALC = alcoholic group.

| Structure | Max | VtxMax | Size (mm²) | X | Y | Z | CWP | CWPLow | CWPHi | NVtxs | Comparison | Contrast |
|---|---|---|---|---|---|---|---|---|---|---|---|---|
| Superior Frontal (RH) | 4.346 | 132708 | 151.42 | 8 | 46.6 | 43.5 | 0.00629 | 0.00449 | 0.00808 | 243 | NC: Men > Women | aversive |
| Caudal Middle Frontal (RH) | 4.129 | 37195 | 128.86 | 38.6 | 9.5 | 43.7 | 0.01641 | 0.01374 | 0.01937 | 191 | NC: Men > Women | aversive |
| Supramarginal (RH) | 4.118 | 74592 | 112.55 | 54.4 | −28.8 | 41.4 | 0.0344 | 0.03029 | 0.0385 | 283 | NC: Men > Women | aversive |
| Rostral Middle Frontal (RH) | −4.624 | 42522 | 108.77 | 22.9 | 54.4 | 17 | 0.03381 | 0.03 | 0.03791 | 144 | Group x Gender | aversive |
| Caudal Middle Frontal (LH) | −6.084 | 47079 | 106.95 | −27.3 | 21.3 | 36 | 0.04462 | 0.03996 | 0.04927 | 230 | Group x Gender | aversive |
| Inferior Parietal (LH) | −3.86 | 112079 | 107.78 | −28.8 | −65.7 | 40.3 | 0.04287 | 0.0385 | 0.04724 | 193 | Men: ALC < NC | happy |

DOI: https://doi.org/10.7554/eLife.41723.030

