## [Decision Letter]

Thank you for submitting your article "Alcoholism Gender Differences in Brain Responsivity to Emotional Stimuli" for consideration by *eLife*. Your article has been reviewed by three peer reviewers, a Reviewing Editor and Joshua Gold as the Senior Editor.

The reviewers have discussed the reviews with one another. The content of each review and the resulting discussion have been integrated into this decision letter to help you prepare a revised submission.

Summary:

This manuscript investigates potential gender differences in AUD-associated alterations to neural correlates of emotion processing. The authors examined fMRI responses in groups of alcohol dependent (ALC) and non-alcohol dependent (NC) individuals, and specifically compared male and female participants. Using a whole-brain analysis, the authors identified several brain regions in which ALC males appeared to have lower contrast activation compared to ALC females and NC males, whereas activation was generally higher in ALC, relative to NC women.

Overall, the manuscript is clear and well-written, the methods are generally appropriate, and the background and discussion are appropriate in scope and content.

Essential revisions:

1) Behavioral methods and analysis:

Why were good/bad/neutral ratings employed? What conceptual model(s) or specific hypotheses drove the selection of the task? Were neural responses related to behavioral ratings? Were behavioral ratings related to affective or neuropsych measures presented in Table 1? If behavioral results are to be meaningfully interpreted, the implications of task performance (and group/sex differences in performance) should be given more measured consideration. These issues are at least partially related to the following concerns.

2) Overstatement/over-interpretation of results:

The behavioral task is not, explicitly, either an emotion judgement task or an emotion regulation measure. Similarly, while data reflecting neural responsivity to task stimuli are of substantive import, they offer only indirect measures of emotion processing and/or emotion regulation. This is somewhat problematic, as several of the authors' points in the Discussion suggest otherwise. Examples of this concern follow:

"Our findings support the view that alcohol can be abused in an effort to restore emotional homeostasis either by increasing or decreasing emotional arousal."

"…the present findings also suggest that the patterns of neural reactivity observed in our study may contribute to the onset of multiple diagnostically distinct syndromes."

Concerns related to overstatement/over-interpretation are highlighted by, but not limited to, the above statements. The authors should distinguish between data which are consistent with a particular hypothesis, and data which offer support for a particular hypothesis. Furthermore, if particular hypotheses/conceptual frameworks (e.g., sex-contingent divergence in alcohol use to increase/decrease arousal) will be used to interpret findings, these frameworks should be more explicitly discussed in the background.

3) Lack of hypotheses and need for additional clarity regarding Type 1 corrections:

Although lack of explicit hypotheses and control for Type 1 error are separate issues, they are conceptually related. Lack of priori hypotheses suggest analyses which are largely exploratory and require more stringent/conservative error correction.

As stated, the hypotheses related to sex differences among AUD individuals suggest only that a "different pattern of abnormalities" will be observed. If the extant literature is insufficient to generate more pointed/specific hypotheses, the authors may consider re-conceptualizing the work as an exploratory study, with Type 1 error correction sufficiently liberal that widespread patterns of sex-contingent differences might be better appreciated? This may be particularly appropriate given that the authors did not attempt to hone in on emotion processing centers in the brain or other regions of the brain that are known to be most affected by chronic alcohol use.

Regardless, the number of conducted contrasts and level of alpha correction remain points of substantial concern. There is consensus among reviewers that the current correction methods are substantially more liberal that those often provided in similar analyses. If manuscript revisions retain these methods, the authors should explicitly address their appropriateness. The Eklund et al. (2016) article may provide guidance in addressing these issues. Regardless of the manner in which the authors choose to address the error correction approach, they should endeavor to characterize the effect sizes of their findings, as doing so will substantially improve data interpretation.

---

## [Author Response]

Essential revisions:1) Behavioral methods and analysis:Why were good/bad/neutral ratings employed? What conceptual model(s) or specific hypotheses drove the selection of the task? Were neural responses related to behavioral ratings? Were behavioral ratings related to affective or neuropsych measures presented in Table 1? If behavioral results are to be meaningfully interpreted, the implications of task performance (and group/sex differences in performance) should be given more measured consideration. These issues are at least partially related to the following concerns.

We restructured the Introduction to reflect the conceptual model (Halgren and Marinkovic, 1995) that originally drove the choice of our task paradigm. We now more clearly describe how our aim was to investigate the process of emotional evaluation, which is why we instructed participants to evaluate how the pictures made them feel (*good, bad*, or *neutral*). We added a simple diagram for the model as a supplement to the figure showing our task paradigm (Figure 1—figure supplement 1).

We analyzed the relationship between (a) behavioral ratings to a number of affective and neuropsychological measures presented in Figure 2 and Appendix 1—tables 1 and 2, and (b) brain activation from the clusters with significant group x gender interactions for erotic vs. neutral and aversive vs. neutral contrasts (the most salient contrasts). We found that percentage ratings were not significantly predicted by interactions of group or gender with the neuropsychological assessment measures (VIQ, PIQ, and DMI), nor by interactions with MAACL Anxiety or MAACL Sensation Seeking scores. As we now note in the Results, “the only significant post-hoc group comparison indicated that for the NC group, POMS Depression scores were positively related to evaluation times for *neutral* ratings in the happy condition (95% confidence interval: [62, 157]), whereas they were not for the ALC group (95% confidence interval: [-19, 40]). In other words, the NC participants with higher Depression scores were slower in rating happy stimuli as being *neutral*.”

Additionally, we found that neither percentage ratings nor evaluation times were significantly related to interactions of group or gender with the contrast effect sizes obtained from the “limbic structures” cluster identified from the erotic contrast. For the aversive vs. neutral contrast, we found significant interactions related to percentage ratings for two clusters, and we added the following text to the Results: “For the “caudal middle frontal cluster 1” and “superior frontal cluster” obtained through analysis of the aversive contrast, percentage ratings were significantly predicted by the interaction of group x gender x rating x contrast effect size. […] That is, while we identified a different pattern in the relationships of percentage ratings to brain activity among the four subgroups, it was not clear how these relationships differed between the ALC_W_ vs. NC_W_, and ALC_M_ vs. NC_M_.” Evaluation times were not significantly related to interactions of cluster contrast effect size with group or gender.

2) Overstatement/over-interpretation of results:The behavioral task is not, explicitly, either an emotion judgement task or an emotion regulation measure. Similarly, while data reflecting neural responsivity to task stimuli are of substantive import, they offer only indirect measures of emotion processing and/or emotion regulation. This is somewhat problematic, as several of the authors' points in the Discussion suggest otherwise. Examples of this concern follow:"Our findings support the view that alcohol can be abused in an effort to restore emotional homeostasis either by increasing or decreasing emotional arousal.""…the present findings also suggest that the patterns of neural reactivity observed in our study may contribute to the onset of multiple diagnostically distinct syndromes."Concerns related to overstatement/over-interpretation are highlighted by, but not limited to, the above statements. The authors should distinguish between data which are consistent with a particular hypothesis, and data which offer support for a particular hypothesis. Furthermore, if particular hypotheses/conceptual frameworks (e.g., sex-contingent divergence in alcohol use to increase/decrease arousal) will be used to interpret findings, these frameworks should be more explicitly discussed in the background.

We have revised the Introduction to include (i) clearly stated hypotheses within a more distinct framework, and (ii) the reviewers’ point about our task being an indirect measure of emotion processing and/or regulation. Similarly, we altered the Discussion considerably, to limit our interpretations, and to make our interpretations more consistent with the data.

3) Lack of hypotheses and need for additional clarity regarding Type 1 corrections:Although lack of explicit hypotheses and control for Type 1 error are separate issues, they are conceptually related. Lack of priori hypotheses suggest analyses which are largely exploratory and require more stringent/conservative error correction.As stated, the hypotheses related to sex differences among AUD individuals suggest only that a "different pattern of abnormalities" will be observed. If the extant literature is insufficient to generate more pointed/specific hypotheses, the authors may consider re-conceptualizing the work as an exploratory study, with Type 1 error correction sufficiently liberal that widespread patterns of sex-contingent differences might be better appreciated? This may be particularly appropriate given that the authors did not attempt to hone in on emotion processing centers in the brain or other regions of the brain that are known to be most affected by chronic alcohol use.Regardless, the number of conducted contrasts and level of alpha correction remain points of substantial concern. There is consensus among reviewers that the current correction methods are substantially more liberal that those often provided in similar analyses. If manuscript revisions retain these methods, the authors should explicitly address their appropriateness. The Eklund et al. (2016) article may provide guidance in addressing these issues. Regardless of the manner in which the authors choose to address the error correction approach, they should endeavor to characterize the effect sizes of their findings, as doing so will substantially improve data interpretation.

We adopted the reviewers’ suggestion of “re-conceptualizing the work as an exploratory study, with Type 1 error correction sufficiently liberal that widespread patterns of sex-contingent differences might be better appreciated.” The manuscript now reflects this re-conceptualization. In the revised manuscript, we note the exploratory nature of the study, and we explain the “liberal” (*p* < 0.05) Type 1 error correction. Also, as described above, we restructured the Introduction around the model that drove the selection of the emotional evaluation task, and made three further changes to the manuscript in accordance with these suggestions.

1) Beginning with the Introduction, we now state that the study was exploratory, and we clarified our general hypotheses. As indicated by our model of Emotion Evaluation and Integration (Figure 1—figure supplement 1), the corticolimbic circuitry is complex and wide ranging, and since our study was exploratory, we did not hypothesize about specific brain regions for gender differences in abnormal activation patterns.

2) We consulted with Howard Cabral, Director of the Biostatistics and Research Design Program of the Boston University Clinical and Translational Sciences Institute. In accordance with the reviewers’ suggestions, he recommended that, given the exploratory nature of the study, the Type 1 error correction of *p* < 0.05 was appropriate. However, we recognize that the multiple hypothesis testing is not something to be ignored, and acknowledge this consideration in the Limitations section. That said, the magnitude of the gender differences are striking, have scientific merit, make sense within our model, and should be shared so they can be followed up by further research. We continue to report the sizes of the effects and characterize the relevance of these sizes in differences in the brain activity.

3) We investigated the cluster correction algorithm we specified using the FS-FAST software with respect to Eklund et al. (2016). We consulted with Doug Greve, the principal creator of the FS-FAST software. He indicated that our use of the default cortical surface multiple comparison correction procedure with thresholds of *p*<0.05 (corrected to *p*<0.0166 for the analysis of three spaces) was appropriate for an exploratory study, given that we additionally provide results using a cluster-forming threshold of *p*<0.001. We now provide these results and describe them in the Limitations section.